# Enhanced thermal stability enables human mismatch-specific thymine–DNA glycosylase to catalyse futile DNA repair

Diana Manapkyzy[1,2], Botagoz Joldybayeva[1,2], Alexander A. Ishchenko[3], Bakhyt T. Matkarimov[4], Dmitry O. Zharkov[5,6], Sabira Taipakova[1,2,4]\*, Murat K. Saparbaev[3]\*

**1** Department of Molecular Biology and Genetics, Faculty of Biology and Biotechnology, al-Farabi Kazakh National University, Almaty, Kazakhstan, **2** Scientific Research Institute of Biology and Biotechnology Problems, al-Farabi Kazakh National University, Almaty, Kazakhstan, **3** Group «Mechanisms of DNA Repair and Carcinogenesis», CNRS UMR9019, Université Paris-Saclay, Gustave Roussy Cancer Campus, Villejuif Cedex, France, **4** National Laboratory Astana, Nazarbayev University, Astana, Kazakhstan, **5** SB RAS Institute of Chemical Biology and Fundamental Medicine, Novosibirsk, Russia, **6** Department of Natural Sciences, Novosibirsk State University, Novosibirsk, Russia

\* murat.saparbaev@gustaveroussy.fr (MKS); sabira.taipakova@gmail.com (ST)

**Data Availability Statement:** All relevant data are within the manuscript and its Supporting Information files with raw data including: - The

## Abstract

Human thymine-DNA glycosylase (TDG) excises T mispaired with G in a CpG context to initiate the base excision repair (BER) pathway. TDG is also involved in epigenetic regulation of gene expression by participating in active DNA demethylation. Here we demonstrate that under extended incubation time the full-length TDG (TDG$^{FL}$), but neither its isolated catalytic domain (TDG$^{cat}$) nor methyl-CpG binding domain-containing protein 4 (MBD4) DNA glycosylase, exhibits significant excision activity towards T and C in regular non-damaged DNA duplex in TpG/CpA and CpG/CpG contexts. Time course of the cleavage product accumulation under single-turnover conditions shows that the apparent rate constant for TDG$^{FL}$-catalysed excision of T from T•A base pairs (0.0014–0.0069 min$^{-1}$) is 85–330-fold lower than for the excision of T from T•G mispairs (0.47–0.61 min$^{-1}$). Unexpectedly, TDG$^{FL}$, but not TDG$^{cat}$, exhibits prolonged enzyme survival at 37°C when incubated in the presence of equimolar concentrations of a non-specific DNA duplex, suggesting that the disordered N- and C-terminal domains of TDG can interact with DNA and stabilize the overall conformation of the protein. Notably, TDG$^{FL}$ was able to excise 5-hydroxymethylcytosine (5hmC), but not 5-methylcytosine residues from duplex DNA with the efficiency that could be physiologically relevant in post-mitotic cells. Our findings demonstrate that, under the experimental conditions used, TDG catalyses sequence context-dependent removal of T, C and 5hmC residues from regular DNA duplexes. We propose that *in vivo* the TDG-initiated futile DNA BER may lead to formation of persistent single-strand breaks in non-methylated or hydroxymethylated chromatin regions.

values behind the means, standard deviations and other measures reported; - The values used to build graphs; - The points extracted from images for analysis.

**Funding:** This work was supported by grants from the Committee of Science of the Ministry of Science and Higher Education of the Republic of Kazakhstan grants  13067762 to S.T. and AP19676334 to S.T. and B.T.M.; French National Research Agency (ANR-22-CE12-0034-01) and Electricité de France RB 2021-05 to M.S.; Fondation ARC PJA-2021060003796 to A.A.I.; Russian Ministry of Higher Education and Science [FSUS-2020-0035] to D.O.Z.; D.M. was supported by fellowship Abai-Vern, Kazakhstan. The funders had no role in study design, data collection and analysis, decision to publish, or preparation of the manuscript. There was no additional external funding received for this study.

**Competing interests:** The authors have declared that no competing interests exist.

**Abbreviations:** 5caC, 5-carboxylcytosine; 5fC, 5-formylcytosine; 5hmC, 5-hydroxymethylcytosine; 5mC, 5-methylcytosine; 8-oxoA, 7,8-dihydro-8-oxoadenine; ANPG, human alkylpurine–DNA glycosylase; AP, apurinic/apyrimidinic; APE1, major human AP endonuclease 1; BER, base excision repair; dmbDNA, dumbbell-shaped duplex oligonucleotide; Hx, hypoxanthine; MBD4, methyl-CpG binding domain-containing protein 4; Nfo, *E. coli* endonuclease IV; TDG, human mismatch-specific thymine–DNA glycosylase; TDG$^{FL}$, native full-length TDG; TDG$^{cat}$, truncated catalytic domain of TDG; TET1–3, ten–eleven translocation family of proteins; U, uracil; UNG, human uracil–DNA glycosylase; Xth, *E. coli* exonuclease III.

## Introduction

Methylation and demethylation of cytosine at the C5 position are essential epigenetic processes in the course of organism development, cell differentiation, genomic imprinting and suppression of mobile elements [1–3]. A major drawback of the epigenetic methylation is that spontaneous deamination of 5-methylcytosine (5mC) generates thymine, resulting in a T•G mismatch. If not repaired, these stray thymines produce C→T mutations at CpG dinucleotides. It is thought that the depleted CpG content in mammalian genomes is due to this high intrinsic mutability of 5mC [4].

In mammalian cells, two DNA repair proteins, methyl-CpG binding domain-containing protein 4 (MBD4 or MED1) and mismatch-specific thymine–DNA glycosylase (TDG), counteract the mutagenic impact of 5mC deamination by removing T from T•G mispairs in a CpG context, followed by reinstallation of a regular C through the base excision repair (BER) pathway [5, 6]. In BER, DNA glycosylases recognize and excise abnormal bases, generating apurinic/apyrimidinic (AP) sites, which are then cleaved by AP endonucleases [7, 8]. While human TDG and MBD4 were first biochemically characterized for their ability to remove T mispaired with G, a more detailed characterization showed that TDG exhibits wider substrate specificity, excising 3,$N^4$-ethenocytosine [9, 10], thymine glycol [11], 5-hydroxycytosine [12], 7,8-dihydro-8-oxoadenine (8-oxoA) [13], mismatched uracil [14] and its derivatives modified at C5 [15]. MBD4 is more selective, excising, beyond T, only uracil, 5-fluorouracil and 5-hydroxymethyluracil opposite to G [6, 16, 17]. Direct comparison of the substrate specificities of these enzymes is though complicated by variable conditions employed in the reported experiments.

In the context of the nucleus, TDG is tightly associated with euchromatin and binds CpG-rich promoters of transcribed genes to scan them for mismatches and to shield them from *de novo* DNA methylation [18, 19]. Another major role of TDG is active epigenetic DNA demethylation. In mammals, Ten–Eleven Translocation family proteins (TETs) oxidize 5mC to 5-hydroxymethylcytosine (5hmC) and then to 5-formylcytosine (5fC) and 5-carboxylcytosine (5caC) [20–23]. TDG efficiently excises 5fC and 5caC nucleobases, but not 5hmC, in a CpG context, suggesting direct involvement of TDG-initiated BER in the active erasure of 5mC during cell reprogramming and embryonic development [23, 24]. TDG knockout mice are embryonic lethal due to the aberrant methylation of promoters of developmental genes and the ensuing failure to establish and maintain cell type-specific gene expression programs [18, 19].

Regulation of transcription via DNA methylation/demethylation read by 5mC-recognizing proteins is a common form of gene control in eukaryotes. Enhancers are *cis*-acting regulatory elements that, when bound by specific transcription factors, boost the activity of their functionally associated genes. Physically, they may be located far away from the transcription start site and regulate gene expression by DNA looping to interact with the target promoters. The function of enhancers can be limited to a particular cell type, time point in the development, or physiological or environmental conditions. It was proposed that promoter and enhancer elements can be activated through generation of transient DNA strand breaks [25]. Recent studies demonstrated that post-mitotic neurons accumulate high levels of persistent single-strand DNA breaks (SSBs) in the enhancers at CpG dinucleotides associated with DNA demethylation that control the expression of genes involved in cell identity [26, 27]. It has been further demonstrated that TET/TDG-mediated removal of 5mC is the source of these persistent SSBs in differentiated neurons and macrophages, suggesting that cycles of DNA methylation/demethylation fuel endogenous DNA damage in post-mitotic cells [28].

Of the products of 5mC oxidation, 5hmC, now often regarded as a "sixth base" of vertebrate DNA, is several orders of magnitude more abundant than 5fC and 5caC [29, 30]. The global genomic content of 5hmC depends on the cell proliferation rate, being the highest in

terminally differentiated cells and lowest in all studied cancers [31, 32]. More than just an intermediate in the demethylation pathway, 5hmC is believed to have its own epigenetic function, since it persists in many cell types, remains stable during cell divisions and is typically enriched in enhancers and bodies of transcriptionally active and tissue-specific genes [33–38]. However, the mechanistic aspects of 5hmC engagement in epigenetic regulation of gene expression remain murky. In mammalian cells, erasure of 5mC and 5hmC marks is thought to be strictly dependent on their further oxidation, since no known DNA glycosylase has been reported to excise these nucleobases [24, 39].

Although the major role of DNA repair is to protect cells from DNA damage, evidence have accumulated showing that mishandling of certain DNA lesions or even normal bases could lead to faulty repair and contribute to cancer and neurodegenerative diseases. For example, human alkylpurine DNA glycosylase (ANPG) can initiate futile BER by removing regular purines from non-damaged DNA [40], and increased level of ANPG is associated with increased risk of lung cancer [41]. Furthermore, we had shown previously that TDG and MBD4 catalyse aberrant removal of T paired with damaged adenine residues [42]. TDG efficiently excises T opposite to hypoxanthine (Hx), 1,$N^6$-ethenoadenine (εA), 8-oxoA and AP site in the TpG/CpX sequence context (where X is a modified residue), while MBD4 removes T from pairs with εA. *In vitro* reconstitution showed that TDG-catalysed aberrant excision of T opposite to Hx initiates repair synthesis forced to use the damaged base as a template, which leads to TpG→CpG and CpA→CpG mutations in a replication-independent manner [42]. These observations suggest that the DNA repair machinery can target regular DNA in an aberrant manner and promote genome instability in the presence of unrepaired DNA lesions.

While the observed aberrant activity of TDG is intriguing, functional characterization of this enzyme has long been plagued by the issue of protein stability *in vitro*. TDG consists of a conserved, well-folded catalytic core and extended, likely disordered N- and C-terminal tails encompassing about a half of the protein's total length. Multiple studies showed that the N-terminal tail is required for efficient binding and excision of T from G•T mispairs and for binding regular DNA [43–46]. Full-length TDG (TDG$^{FL}$, 410 amino acids long) and partially truncated TDG (residues 82–308) are several fold more active than the isolated catalytic domain (residues 111–308, TDG$^{cat}$) [46, 47]. On the other hand, the presence of the intrinsically disordered tails makes TDG quite labile and notably affects its activity and specificity in a temperature-dependent way. This issue was extensively studied by Drohat's group, revealing that at 37˚C TDG is prone to irreversible unfolding and aggregation, while remaining stable for at least 5 h at 22˚C [48]. Guided by these observations, they used 22˚C as the standard conditions to characterize the substrate specificity of TDG, detecting no removal of T paired with A, nor of C, 5mC and 5hmC paired with G even after an extended incubation [12, 24, 49, 50]. At the same time, the G•T-specific activity of TDG depends strongly on the temperature with an 11-fold rise from 5˚C to 37˚C, while the ability to excise uracil increases only threefold at this interval [48]. The excision of 5fC and 5caC opposite to G is also 3–4 fold higher at 37˚C than at 22˚C [24]. Curiously, Drohat and colleagues also observed that the presence of an excess of non-specific DNA stabilized TDG at 37˚C for over 2 h, suggesting that DNA binding somehow prevents the protein aggregation [48].

In our previous studies, when characterizing TDG excision of T paired with a damaged adenine, we serendipitously found that upon long incubation at 37˚C TDG can also excise pyrimidine bases from control normal DNA duplexes. This led us to inquire into the unusual biochemical properties of TDG acting on regular pyrimidines, 5mC and 5hmC in otherwise undamaged DNA.

## Materials and methods

### Proteins

Phage T4 polynucleotide kinase and *E. coli* exonuclease III (Xth) were purchased from New England Biolabs (Ipswich, MA). *E. coli* ArcticExpress (DE3) cells were from Merck Biosciences (Darmstadt, Germany). The purified BER proteins including truncated version of human uracil–DNA glycosylase (UNGΔ84), *E. coli* endonuclease IV (Nfo) and human AP endonuclease 1 (APE1) were from laboratory stocks [51]. Expression and purification of TDG and MBD4 is described below. The activities of DNA glycosylases were verified using their canonical substrates immediately before use.

### Oligonucleotides

Sequences of the DNA and RNA oligonucleotides used in this work are shown in Table 1. All oligonucleotides were synthesized by Eurogentec (Seraing, Belgium). To follow enzymatic cleavage, the oligonucleotides were 5′-end labelled with $\gamma[^{32}P]$-ATP (PerkinElmer, Shelton, CT) or carried a fluorescent dye Cy5 conjugated to the 5′-end. To obtain duplex substrates, they were annealed with corresponding complementary strands as described previously [52]. The resulting duplexes are referred to as X*•N, where X is a residue in the labelled strand and N is a base opposite to X in the complementary non-labelled strand. It should be noted that the majority of data in this work were obtained with $^{32}P$-labelled DNA substrates, while both Cy5-labelled and radioactively labelled oligonucleotides were used to perform kinetic measurements.

### Expression and purification of TDG and MBD4

His-tagged human $TDG^{FL}$ (wild-type and the inactive N140A mutant), $TDG^{cat}$ and MBD4 proteins were produced as described previously [13, 51]. Briefly, *E. coli* ArcticExpress (DE3) cells were transformed with the expression vectors pET28c-TDG, pET28c-$TDG^{N140A}$, pET28c-$TDG^{cat}$ or pET6H-MBD4 and grown with vigorous shaking at 37˚C in LB medium supplemented with 50 μg/ml of kanamycin or ampicillin to $OD_{600}$ = 0.6–0.8. Then the temperature was reduced to 12˚C, the expression was induced by 0.2 mM isopropyl-β-D-1-thiogalactopyranoside, and the cells were further grown for 15 h. All purification procedures were carried out at 4˚C. The bacteria were harvested by centrifugation and the cell pellets were lysed using a French press at 18,000 psi in a buffer containing 20 mM HEPES–KOH (pH 7.6), 40 mM NaCl and 0.025% Nonidet P-40 supplemented with Complete$^{TM}$ protease inhibitor cocktail (Roche Diagnostics, Basel, Switzerland). The lysates were cleared by centrifugation at 40,000×*g* for 1 h at 4˚C, the supernatant was adjusted to 500 mM NaCl and 20 mM imidazole and loaded onto a HiTrap Chelating HP column (GE Healthcare, Chicago, IL). The column was washed with Buffer A1 consisting of with Buffer A1 consisting of 20 mM HEPES–KOH pH 7.6), 500 mM NaCl, 20 mM imidazole, and the bound proteins were eluted with a linear gradient of 20–500 mM imidazole in Buffer B1(20 mM HEPES–KOH pH 7.6), 500 mM NaCl, 500 mM imidazole). The fractions containing the proteins of interest were pooled, diluted tenfold with Buffer A2 consisting of 20 mM HEPES–KOH pH 7.6), 50 mM NaCl and 0,0125% Nonidet P-40 and loaded to a 1-ml HiTrap Heparin column (GE Healthcare, Chicago, IL). Proteins bound to the column were eluted with a 50–800 mM NaCl gradient in Buffer B2 (20 mM HEPES–KOH pH 7.6), 1000 mM NaCl and 0,0125% Nonidet P-40). The eluted fractions were analysed by polyacrylamide gel electrophoresis (PAGE) in the presence of sodium dodecyl sulfate (SDS), and the fractions containing the pure TDG, $TDG^{cat}$ and MBD4 proteins were stored at −80˚C in 50% glycerol. The concentration of purified proteins was determined by the Bradford assay.

**Table 1. Sequences of the oligonucleotides used in the study.**

| ID [a] | Length | Sequence (5′→3′) [b] | Comment |
|---|---|---|---|
| | | *Oligodeoxyribonucleotides* | |
| 14–03 | 27 | CCATCATCTCCAGAC**R**GATCCTCACAC | Contains A or G (**R**) at position 16 in CpR context [c] [53]. |
| c14-03 | 27 | GTGTGAGGATC**T**GTCTGGAGATGATGG | Complementary to 14–03, places **T** in a TpG context opposite **R** [53]. |
| 55-mer hairpin | 55 | ggtacgttcgtacc GTGTGAGGATC**T**GTCTGGAGATGATGG gccgtcttgacggc | Forms 6-bp hairpins at both ends (lowercase) exposing a 27-nt single-stranded gap complementary to 14–03; places **T** in a TpG context opposite **R** [54]. |
| 14-03-U11 | 27 | CCATCATCTC**U**AGACAGATCCTCACAC | Contains U at position 11 in a CpU context. |
| 10–05 | 24 | TCTTCTTCTGTGC**R**CTCTTCTTCT | Contains A or G (**R**) at position 14 in a CpR context [55]. |
| c10-05 | 24 | AGAAGAAGAG**T**GCACAGAAGAAGA | Complementary to 10–05, places **T** in a TpG context opposite **R** [55]. |
| 10–13 | 24 | TCTTCTTCT**R**TGCACTCTTCTTCT | Same as 10–05, but contains A or G (**R**) at position 10 in a TpA context [55]. |
| c10-13 | 24 | AGAAGAAGAG**T**GCATAGAAGAAGA | Complementary to 10–13, places **T** in a TpG context opposite **R** [55]. |
| 63 | 63 | ACAGCACCAGATTCAGCAATTAAGCTCTAAG**Y**CATCCGCAAAAATGACCTCTTATCAAAAGGA | Contains T, C or U (**Y**) at position 32 in a GpY context [54]. |
| c63 | 63 | TCCTTTTGATAAGAGGTCATTTTTGCGGATG**R**CTTAGAGCTTAATTGCTGAATCTGGTGCTGT | Complementary to 63, places A or G (**R**) in an RpC context opposite **Y** [54]. |
| mm876-Enhan | 63 | AACAACAAAGGACGGTTAGACGTTCCAGCTGGTATGAAACAGACAAGAGTCGGCAGAGGAGGC | Enhancer sequence of mouse element mm876 [56]. |
| c. mm876-Enhan | 63 | GCCTCCTCTGCCGACTCTTGTCTGTTTCATACCAGCTGGAACGTCTAACCGTCCTTTGTTGTT | Complementary to mm876-Enhan [56]. |
| 28.Adr | 28 | GGGAGAAGAGGAGGAA**T**GAAGAGAGCTC | Contains **T** at position 17 in a single TpG context [50]. |
| c28.Adr | 28 | GAGCTCTCTTC**A**TTCCTCCTCTTCTCCC | Complementary to 28.Adr, places **A** in a single CpA context opposite **T** [50]. |
| 19.AD | 19 | GCTCTGTACATGAGCAGTG | Contains TpG, CpA, and TpA [49]. |
| c19.AD | 19 | CACTGCTCATGTACAGAGC | Complementary to 19.AD [49]. |
| RARE | 27 | CGCCGGGTTCACGG**C**GGGGTCAGCGGC | Enhancer for retinoic acid receptor, contains C at position 15 in a CpG context [57]. |
| cRARE | 27 | GCCGCTGACCCCGCCGTGAACCCGGCG | Complementary to RARE [57]. |
| RARE-mC15 | 27 | CGCCGGGTTCACGG**M**GGGGTCAGCGGC | Same as RARE but contains 5mC (**M**) at position 15 in a CpG context [57]. |
| cRARE-mC12 | 27 | GCCGCTGACCC**M**GCCGTGAACCCGGCG | Complementary to RARE but contains 5mC (**M**) at position 12 in a CpG context [57]. |
| RARE-hmC15 | 27 | CGCCGGGTTCACGG**H**GGGGTCAGCGGC | Same as RARE, but contains 5hmC (**H**) at position 15 in a CpG context [57]. |
| cRARE-hmC12 | 27 | GCCGCTGACCC**H**GCCGTGAACCCGGCG | Complementary to RARE, but contains 5hmC (**H**) at position 12 in a CpG context [57]. |
| 14.CpG-C16 | 27 | TAGACTTCGTCGACT**C**GACTTCGAGCT | Sequence rich in CpG, contains **C** at position 16 in a CpG context. |
| c14.CpG-C16 | 27 | AGCTCGAAGT**C**GAGTCGACGAAGTCTA | Complementary to 14.CpG-C16. Contains **C** at position 11 in a CpG context. |
| 14.CpG-mC16 | 27 | TAGACTTCGTCGACT**M**GACTTCGAGCT | Same as 14.CpG but contains 5mC (**M**) at position 16 in a CpG context. |
| 14.CpG-hmC16 | 27 | TAGACTTCGTCGACT**H**GACTTCGAGCT | Same as 14.CpG, but contains 5hmC (**H**) at position 16 in a CpG context. |

(*Continued*)

**Table 1.** (Continued)

| ID [a] | Length | Sequence (5′→3′) [b] | Comment |
|---|---|---|---|
| *Oligoribonucleotides* | | | |
| GC-RNA | 32 | GCGCGCGCGCGCGCGAGAGCGCGCGCGCGCGC | Forms a hairpin structure [58]. |
| Mut-RNA | 36 | GCGCGCGCGCGCGCGAGAGCGCGCGAAAACGCGGGAGG | Forms secondary structures [58]. |
| A30-RNA | 30 | AAAAAAAAAAAAAAAAAAAAAAAAAAAAAA | Single-stranded homo-oligoA [58]. |

[a] To aid comparison with the data in the literature, the naming of oligonucleotides, when applicable, is kept consistent with previously published studies (referenced in the Comment column) in which these sequences were used.

[b] R, A or G; Y, C, T or U; M, 5mC; H, 5hmC.

[c] The relevant nucleotide context is underlined.

## DNA repair activity assays

Unless indicated otherwise, the standard reaction mixture (20 µl) for DNA repair assays contained 20 nM $^{32}$P-labelled oligonucleotide duplex, 20 mM Tris–HCl (pH 8.0), 1 mM ethylenediaminetetraacetate (EDTA), 1 mM dithiothreitol (DTT), 100 µg/ml bovine serum albumin (BSA) and 200 nM TDG$^{FL}$, TDG$^{cat}$ or MBD4. The reaction mixtures were incubated at 37˚C for 5 to 60 min when measuring TDG-specific activities (T*•G and U*•G duplexes; the asterisk denotes the labelled strand containing the specified base in a defined position) or 1 h and 18 h when measuring TDG-catalysed futile activity (T*•A, T•A*, C*•G and C•G* duplexes). The reaction mixture for Nfo contained 50 mM KCl, 20 mM HEPES–KOH (pH 7.6), 100 µg/ml BSA, 1 mM DTT and 1 nM Nfo. The reaction for Xth was performed in NEBuffer 1 (New England Biolabs) with 1 nM enzyme. The reaction mixture for UNGΔ84 contained 20 mM Tris–HCl (pH 8.1), 100 mM NaCl, 1 mM EDTA, 1 mM DTT, 100 µg/ml BSA, and was incubated for 30 min at 37˚C. All reactions were stopped by adding 10 µl of a stop solution consisting of 0.5% SDS and 20 mM EDTA. After incubation, unless indicated otherwise, the samples were treated with 0.1 M NaOH for 3 min at 95˚C to cleave AP sites left after base excision by DNA glycosylases, after which the mixtures were neutralized with HCl. In some experiments the samples were treated either with light piperidine (10% v/v piperidine at 37˚C for 30 min) or with 10 nM APE1 in TDG buffer supplemented with 5 mM MgCl$_2$. To analyse the reaction products, the samples were desalted using Sephadex G-25 DNA grade SF column (GE Healthcare) equilibrated in 7.5 M urea and separated by electrophoresis in a denaturing 20% (w/v) polyacrylamide gel (7 M urea, 0.5×TBE, 42˚C). The gels were exposed to a Fuji FLA-3000 Phosphor Screen (Fujifilm, Tokyo, Japan), scanned with Typhoon FLA 9500 imager (GE Healthcare) and quantified using Image Gauge v4.0 software (Fujifilm). At least three independent experiments were conducted for all kinetic measurements.

To generate size markers, 20 nM $^{32}$P-labelled duplexes were incubated either with 10 nM Nfo for 5–60 min at 37˚C, or with 1 U of Xth for 10 min at room temperature. The reactions were stopped by adding stop solution (10 mM EDTA, 0.25% SDS) and heating at 95˚C for 5 min. Alternatively, size markers were generated by using Fast Bisulfite Conversion kit (Qiagen, Venlo, the Netherlands) and UNG. For this, 20 nM $^{32}$P-labelled single-stranded oligonucleotides was treated with sodium bisulfite according to the manufacturer's instructions and then incubated with 10 nM UNG at 37˚C for 10 min followed by hot alkaline treatment to cleave at AP sites.

## Single-turnover kinetics

To obtain rate constants ($k_{obs}$) not affected by enzyme–substrate association or by product inhibition, such that $k_{obs}$ reflects the maximal base excision rate ($k_{obs} \approx k_{max}$), we used single-

turnover kinetics under a large excess of the enzyme over the substrate ($[E] >> [S] > K_d$). The time courses were performed in a large volume with 200 nM TDG and 20 nM substrate at 37°C. At each time point, a 20-μl aliquot was withdrawn and treated and analysed as described above. The data were fitted by non-linear regression to one-phase exponential rise to maximum Eq S1 using GraphPad Prism v5 software (GraphPad Software, Boston, MA):

$$[\text{Fraction product}] = A(1 - e^{-k_{\text{obs}}t}) \tag{EqS1}$$

where $A$ is the maximum fraction product, $k_{\text{obs}}$ is the rate constant, and $t$ is the reaction time.

## Bioinformatic analysis

For residue coevolution analysis, sequences of all TDG homologs from chordates were retrieved from the NCBI RefSeq Protein database using human full-length TDG (NP_003202.3) as a query. After excluding incomplete and low-quality entries, the remaining 402 sequences were aligned with Clustal Omega [59] and analysed using CRASP [60] for correlations in physicochemical parameters within amino acid substitutions. Protein disorder and tendency to undergo phase separation was analysed using ParSe v2 [61].

## Molecular modelling

The model of full-length TDG generated by AlphaFold2 (AF-Q13569-F1-model_v2) [62, 63] with unfolded N- and C-tails was subject to coarse-grained Monte Carlo molecular simulation in CABS-flex v2.0 [64, 65]. Restraints (linear potential with slope 1 beyond 3.8–8.0 Å) were applied to pairs in which both residues are located in secondary structure elements (helices or sheets) separated by $> 3$ positions; the temperature factor was set at 1.4. Ten trajectories of 50,000 models were generated; for each, 1,000 models were extracted. After clustering, ten centroid structures for each of the ten trajectories were converted to all-atom representation and analysed using MDTRA [66].

## Results

### Native TDG excises regular pyrimidines from non-damaged duplex DNA and generates abasic sites

Previously, in our unpublished studies, we observed weak activity of native TDG towards regular DNA oligonucleotide duplexes upon long incubation at 37°C, which we tended to attribute to trace contaminating non-specific nucleases. In the present work, we decided to examine whether TDG$^{FL}$ can act on regular DNA substrates using the catalytically inactive TDG$^{FL}$-N140A mutant as a control for the presence of nuclease contamination from the overproducing bacterial cells. Furthermore, to avoid degradation of a DNA substrate by putative nuclease activities, we constructed a 41-mer dumbbell-shaped duplex (dmbDNA) composed of a long 55-mer strand forming hairpins at both ends ("55-mer hairpin", Table 1) and a complementary 27-mer oligonucleotide annealing in the middle (14–03; Fig 1 and Table 1). This type of DNA construct was used in our previous study of APE1 [54]. The 27-mer fragment, which contained either A or G at position 16 or U at position 11, was annealed to the 5′-$^{32}$P-labelled 55-mer containing a regular T at position 26 (i. e., opposite to A$^{16}$ or G$^{16}$ mentioned above), to obtain the dmbDNA substrate in which the A$^{16}$ or G$^{16}$ residues of 14–03 are located in a CpR sequence context (R, purine base; Fig 1B). To examine DNA glycosylase activity on both strands of a duplex, we also used dmbDNA with the 5′-$^{32}$P-labelled 27-mer strand annealed to the non-labelled 55-mer hairpin oligonucleotide. The resulting dmbDNA substrates G16•T*, T•A*, A•T* and T•U11* (asterisk denotes the labelled DNA strand with the residue of interest)

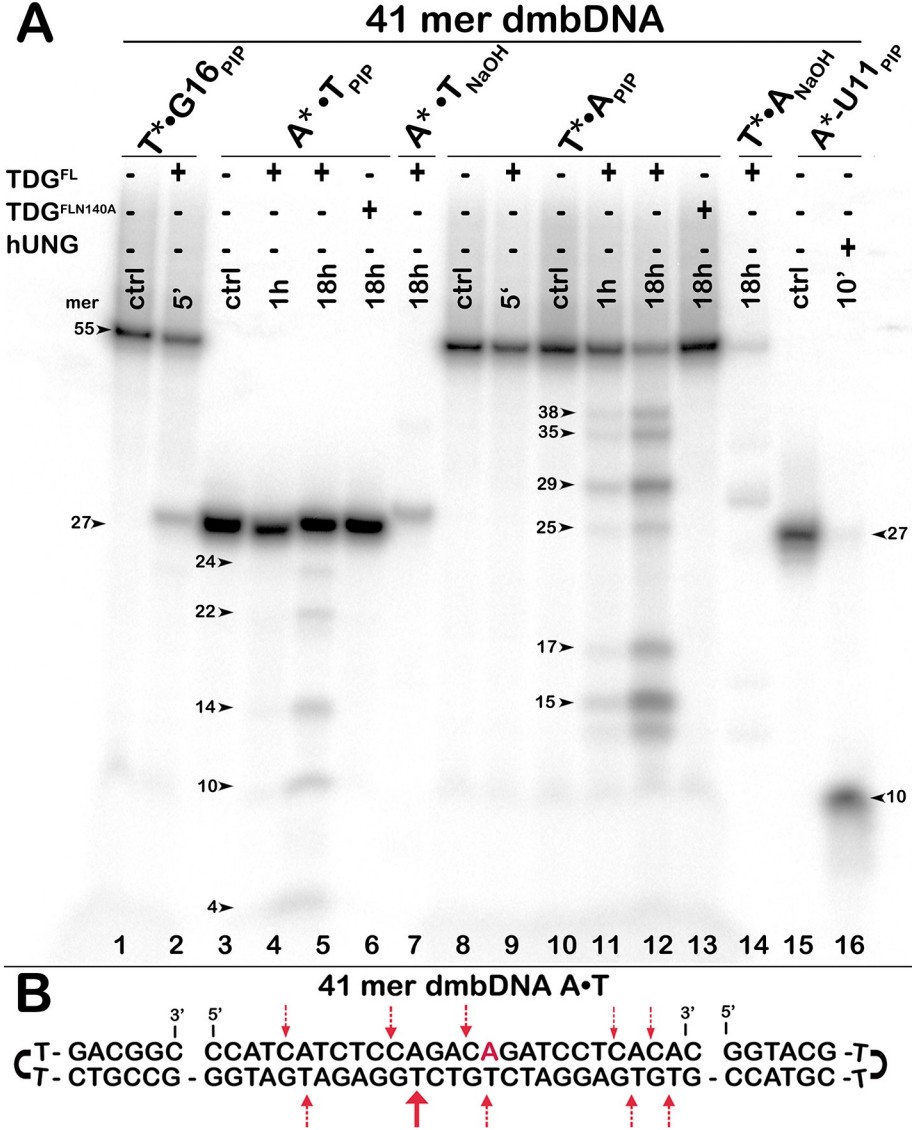

**Fig 1. Action of native TDG$^{FL}$ and catalytically dead mutant TDG$^{FL}$-N140A on regular 41-mer 14–03 A\*•T and A•T\* dmbDNA substrates.** (**A**) Denaturing PAGE analysis of cleavage products. Lane 1, control 41-mer G•T\* dmbDNA containing T at position 26; lane 2, same as lane 1 but treated for 5 min with TDG$^{FL}$. Lanes 3–7, 41-mer A\*•T dmbDNA containing A at position 26 (27-mer top strand labelled): lane 3, no enzyme; lanes 4–5, TDG$^{FL}$, 1 h and 18 h; lane 6, TDG$^{FL}$-N140A, 18 h; lane 7, same as lane 5 but treated with NaOH instead of light piperidine. Lanes 8–14, 41-mer A•T\* dmbDNA containing A at position 26 (55-mer bottom strand labelled): lanes 8 and 10, no enzyme; lanes 9, 11 and 12, TDG$^{FL}$, 5 min, 1 h and 18 h; lane 13, TDG$^{FL}$-N140A, 18 h; lane 14, same as lane 12 but treated with NaOH instead of light piperidine. Lane 15, 5′-$^{32}$P-labelled 27-mer top strand containing U at position 11; lane 16, same as lane 15, but with UNG. Arrows mark the size of the DNA substrate and the cleavage fragments. (**B**) Schematic representation of dmbDNA sequence with arrows pointing to the pyrimidines excised by TDG$^{FL}$. For details, see Materials and Methods.

were incubated in the presence of TDG$^{FL}$ or TDG$^{FL}$-N140A for 1 h or 18 h at 37°C, followed by either light piperidine or hot alkaline treatment to cleave AP site generated by the DNA glycosylase action.

As shown in Fig 1, 5-min incubation of the 41-mer dmbDNA G16•T\* duplex with a limiting amount of TDG$^{FL}$ expectedly resulted in the excision of T opposite to G and the generation

of a 25-mer cleavage product from the 55-mer hairpin (Fig 1A, lane 2). Note that the 25-mer product (Fig 1A, lane 2) migrates slower than the 27-mer strands 14–03 of the A*•T duplex and 14-03-U11 of the T•U11* duplex (Fig 1A, lanes 3 and 15), possibly due to the presence of a hairpin structure in the former. Unexpectedly, long 18-h incubation of TDG$^{FL}$ with A*•T dmbDNA, in which the 27-mer strand 14–03 with A at position 16 is labelled, yielded 24-, 22-, 14-, 10- and 4-mer cleavage fragments (Fig 1A, lane 5), suggesting futile excision of regular C opposite to G in a CpA context (Fig 1B, dashed arrows above the top strand). Moreover, 1-h and 18-h incubation of TDG$^{FL}$ with A•T* dmbDNA, in which the 55-mer strand with T at position 26 was labelled, produced 38-, 35-, 29-, 25-, 17- and 15-mer cleavage fragments of varying intensities (Fig 1A, lanes 11 and 12), implying the excision of regular T residues opposite to A at positions 39, 36, 30, 26, 18 and 16 in the hairpin oligonucleotide (Fig 1B, arrows below the bottom strand). Notably, in the absence of mismatches, TDG$^{FL}$ preferentially excised T from positions 30 and 16 of the 55-mer rather than from position 26 as in the control G16•T* substrate. Note that two samples treated by hot alkali instead of light piperidine were not freshly prepared and contained less radioactivity than the rest of the samples because of $^{32}$P decay (Fig 1A, lanes 7 and 14). Mutant TDG$^{FL}$-N140A cleaved neither A*•T nor A•T* dmbDNA after 18-h incubation at 37˚C (Fig 1A, lanes 6 and 13), indicating that the futile excision of T is not due to a bacterial host contamination but is an intrinsic property of the human DNA glycosylase.

To inquire into the nature of the observed activity, we analysed the reaction products by denaturing PAGE before and after hot alkali treatment. As shown in Fig 2, in the absence of NaOH, incubation of TDG$^{FL}$ with 14–03 A•T* dmbDNA for 18 h produced only faint smeared cleavage fragments (Fig 2, lane 7), confirming that the enzyme cannot directly cleave the DNA backbone. As expected, when TDG$^{FL}$ reaction products were treated by hot alkali, distinct cleavage fragments appeared (Fig 2, lanes 8 and 9), suggesting that the observed products are not of an AP lyase or a nuclease origin, but result from TDG$^{FL}$ excising regular pyrimidines to generate AP sites in duplex DNA. To further substantiate the appearance of AP sites after TDG-catalysed futile excision, we compared the products of several well-established post-treatment methods used to cleave DNA at AP sites. For this we incubated $^{32}$P-labelled dmbDNA G•T* and A•T* substrates with TDG$^{FL}$ for 1 h and 18 h at 37˚C, respectively, and treated the reaction products with hot NaOH, light piperidine, or APE1. Consistent with the respective mechanisms of AP site cleavage, hot NaOH generated a fast-migrating 31-mer fragment with a 3′-phosphate terminus, light piperidine treatment yielded the slowest 31-mer fragment with 3′-terminal α,β-unsaturated aldehyde residue, and APE1 produced an intermediate-mobility 31-mer cleavage fragment with a 3′-OH end (S1 Fig). We observed no significant difference in the quantity of the products between the treatments, indicating that TDG demonstrates G/T-specific and futile activities irrespective of the method used to reveal the nascent AP sites (S1 Fig). Strikingly, TDG$^{cat}$ (residues 111–308) exhibited dramatically lower futile activity towards 14–03 A•T* dmbDNA than did its full-length counterpart (Fig 2, lane 5 vs lane 9). After 18 h, TDG$^{cat}$ excised only 0.5–0.8% of T at positions 39, 36, 30, 26, 18 and 16 in the 55-mer strand of 14–03 A•T* dmbDNA, whereas TDG$^{FL}$ excised 3.3–30.1% of T at the same positions. These results suggest a possible role of non-catalytic N-terminal (residues 1–110) and C-terminal (residues 309–410) tails of human TDG in the futile DNA excision.

In addition, we examined whether MBD4, a functional analog of TDG, is capable of similar futile T excision from DNA duplexes. We incubated human MBD4 with G•T* and A•T* duplexes for 1 h and 18 h, respectively. As expected, MBD4 efficiently excised mismatched T opposite to G but failed to exhibit detectable activity towards a regular DNA duplex (S2 Fig),

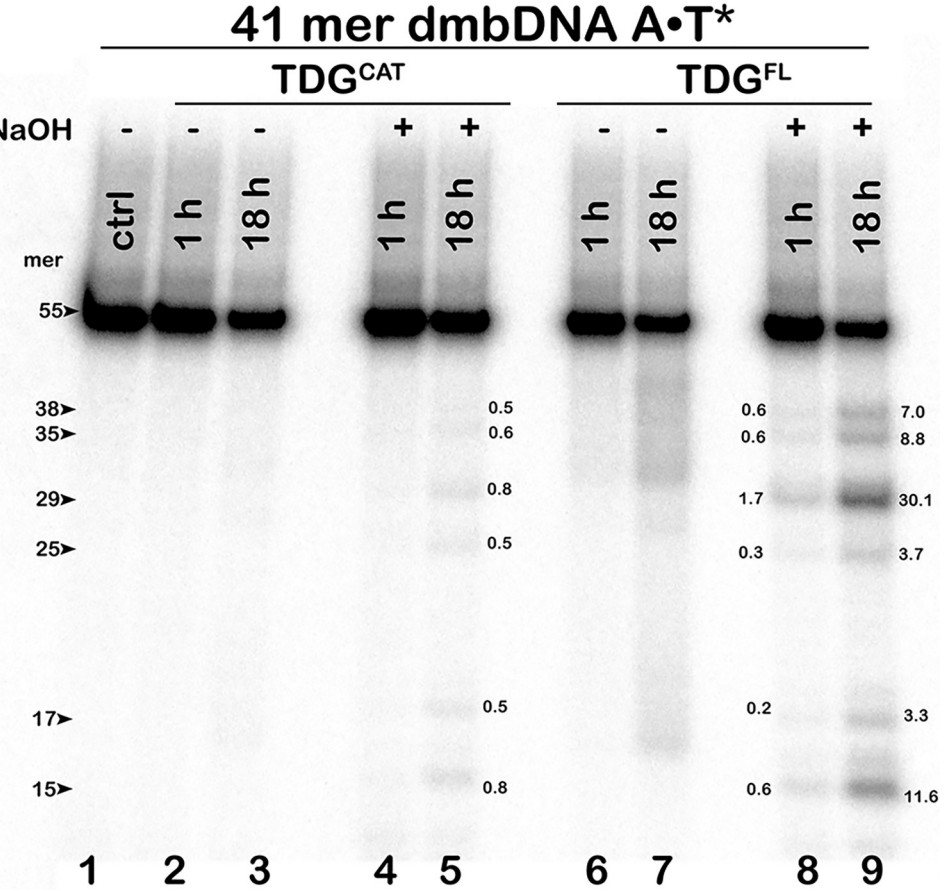

**Fig 2. Denaturing PAGE analysis of the cleavage products following the action of TDG^cat and TDG^FL on the regular 41-mer 14–03 A•T\* dmbDNA substrate.** After the reaction, samples were either treated or not with hot alkali. Lane 1: control 14–03 dmbDNA with 5′-^32P-labelled bottom 55-mer strand; lanes 2–3, same as lane 1 but with TDG^cat, no hot alkali treatment; lanes 4–5, same as lanes 2–3, but with hot alkali treatment; lanes 6–9, same as lanes 2–5 but with TDG^FL. Arrows mark the size of the DNA substrate and the cleavage fragments. The numbers next to the bands in lanes 5, 8 and 9 correspond to the percentage of cleavage products. For details, see Materials and Methods.

suggesting the difference in the mechanism of substrate recognition between these G/T-specific DNA glycosylases.

## Kinetics of TDG^FL-catalysed futile thymine excision

We further characterized the substrate specificity of TDG^FL by comparing the kinetic parameters of cleavage of short blunt-ended 27-mer 14–03 and 24-mer 10–05 oligonucleotide duplexes containing either G•T\* or A•T\* base pair at defined positions in a TpG sequence context. Reactions were performed under single-turnover conditions using a 10-fold molar excess of the enzyme over the substrate, which provides the maximal rate of base excision ($k_{obs}$) for a given substrate. As anticipated, product accumulation time courses showed that G•T is preferred by TDG^FL; >80% of T at position 16 in the mismatched 14–03 G•T\* duplex was excised already in 5 min at 37˚C (Fig 3A and 3B) in contrast to only 21% and 18% of T at positions 16 and 12, respectively, in a regular 14–03 A•T\* duplex after 18 h at 37˚C (Fig 3C and 3D). Similar large differences between the conventional and futile DNA glycosylase activities were observed in the kinetics of excision of T by TDG^FL from 10–05 G•T\* and A•T\* duplexes (S3 Fig). From these time courses, we calculated $k_{obs}$ values for TDG-catalysed

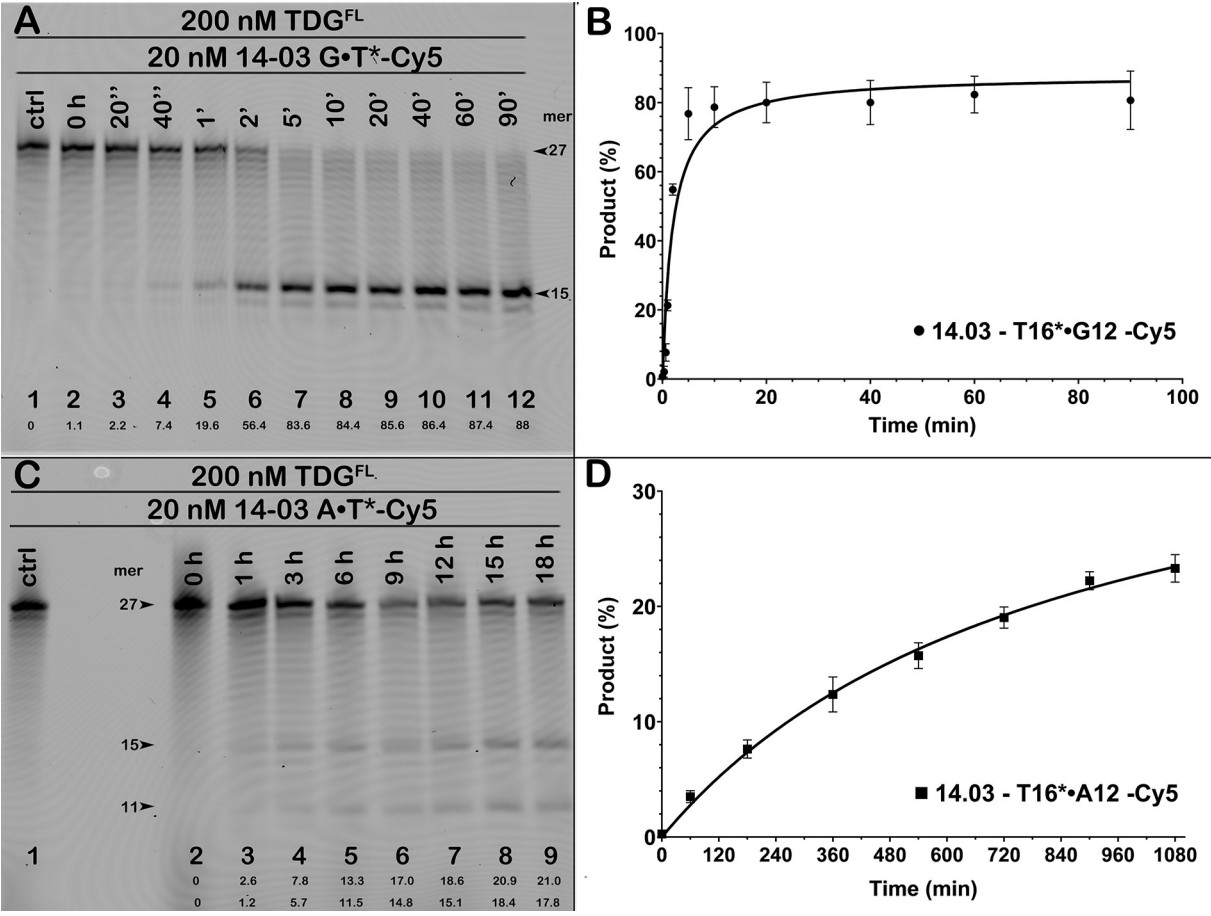

**Fig 3. Time courses of TDG^FL-catalysed excision of T from G•T and A•T base pairs in the 27-mer 14–03 duplex. (A, C)** Denaturing PAGE analysis of the reaction products. Time courses were performed using 200 nM TDG and 20 nM 5′-Cy5-labelled 27-mer 14–03 duplex containing G•T or A•T base pairs at position 16. Arrows mark the size of the DNA substrate and the cleavage fragments. Percentage of cleavage products is indicated under the gel images. **(B, D)** Plots of pre-steady-state single turnover kinetic of TDG^FL-catalysed cleavage of 14–03 duplexes. Mean ± SD of three independent experiments is shown. For details, see Materials and Methods.

canonical and futile activities. As shown in Table 2, when using the 14–03 duplex as a DNA substrate, the $k_{obs}$ values of TDG^FL-catalysed excision of T opposite to A were, respectively, 330- and 270-fold lower than that of mismatched T (0.47 min$^{-1}$). When comparing the $k_{obs}$

**Table 2. Pre-steady-state kinetic parameters of TDG-catalysed excision of T opposite to A or G in regular DNA duplexes.**

| Substrate [a] | $k_{obs}$ (min$^{-1}$) | $k_{obs[G]}/k_{obs[A]}$ |
|---|---|---|
| 14–03 T16/G12, G•T* (TpG/CpG) | 0.47 ± 0.09 | n/a [b] |
| 14–03 T12/A16, A•T* (TpG/CpA) | 0.0014 ± 0.0006 | 330 |
| 14–03 T16/A12, A•T* (TpG/CpA) | 0.0018 ± 0.0010 | 270 |
| 10–05 G14/T11, G•T* (TpG/CpG) | 0.61 ± 0.16 | n/a |
| 10–05 A14/T11, A•T* (TpG/CpA) | 0.0060 ± 0.0041 | 100 |
| 10–05 T11/G14, T*•G (TpG/CpG) | 0.59 ± 0.14 | n/a |
| 10–05 T11/A14, T*•A (TpG/CpA) | 0.0069 ± 0.0029 | 85 |

[a] The sequence context is given in the parentheses.

[b] N/a, not applicable for T•G.

values obtained on 10–05 DNA duplexes, the futile activity towards A•T base pair was 2.5–3-fold greater than for the 14–03 duplex while the activity on a mismatched T was only 1.3-fold higher, resulting in 85- and 100-fold differences between the canonical and futile activities on 10–05 substrates (Table 2). These results suggest that the efficiency of both mismatched and normal base excision by TDG$^{FL}$ markedly depends on the DNA sequence context. Moreover, given a huge excess of normal DNA in the chromatin, the measured rate constants are consistent with the possibility of futile action of TDG *in vivo*.

Previously Morgan *et al*. reported that TDG excises T from a 19-mer T*•A duplex with an exceedingly low rate, ~17,600-fold slower than the activity on a T*•G duplex ($k_{max} = 1.3 \times 10^{-5}$ min$^{-1}$ *vs* 0.22 min$^{-1}$) [49]. In a later study by Maiti *et al*. from the same laboratory, the futile activity of TDG was measured using a $^{32}$P-labelled 28-mer T*•A duplex and again observed a ~20,000-fold kinetic preference in favour of mismatched T excision [50]. As noted in the Introduction, in order to avoid enzyme inactivation over the long time course obligatory for futile activity measurements, the reaction was run at 22˚C in these studies, which may explain the deviation from only 100-fold preference for T*•G *vs* T*•A substrates observed in our work. It should also be noted that the 28-mer duplex used in [50] had a very particular sequence context with the cleaved strand highly enriched in purines with a single T in a TpG context.

To benchmark our results directly against the published data, we examined the futile activity of TDG$^{FL}$ on $^{32}$P-labelled 19-mer and 28-mer A•T* duplexes (19AD and 28Adr, respectively; Table 1) used in [49, 50] at 22˚C and 37˚C under our experimental conditions (S4 and S5 Figs). As expected, after 18 h at 37˚C, TDG$^{FL}$ was able to excise 3–10% of T from A•T base pairs at several positions in the top and bottom strands of 19AD/c19AD and 28Adr/c28Adr duplexes (S4 Fig), indicating that TDG exhibits futile activity in a wide range of sequence contexts. Treatment of a 27-mer 14–03 G•T* duplex with TDG$^{FL}$ at 22˚C and 37˚C for 1 h did not show significant difference in the excision of mismatched T (S5A Fig). On the contrary, incubation of a 27-mer 14–03 A•T* duplex with TDG$^{FL}$ at 22˚C and 37˚C for 18 h revealed a ~5-fold lower excision of T opposite to A at 22˚C (S5A Fig). When TDG$^{FL}$ was presented with 19AD and 28Adr duplexes in which either strand was labelled, we observed >10-fold less excision of T opposite to A at 22˚C than at 37˚C (S5B Fig). The futile activity measured on the same 28Adr duplex at 37˚C here and at 22˚C in [50] revealed more than an order of magnitude difference: 9.9% of T excised in 1080 min (18 h) at 37˚C *vs* 3% in 4000 min (66.7 h) at 22˚C (S4 Fig). Yet at 22˚C little if any (~1.4-fold) difference was observed between our data and [50] on 28Adr: 0.6% T excised in 1080 min here (S5B Fig) *vs* 3% in 4000 min in [50]. Taken together, these results suggest that the nature of the enzyme (full-length or truncated), DNA substrate sequence contexts, reaction temperature and incubation time are critical for the detection and characterization of the futile activity of TDG.

Long-term stability of TDG

Time courses shown in Fig 3 and S2 Fig indicate that, even after 6, 9 and 12 h at 37˚C, TDG$^{FL}$ continues to excise T from a regular DNA duplex, suggesting that the enzyme somehow maintains its activity despite the problem of long-term stability. Native TDG contains largely disordered N- and C-terminal tails [46, 67], which were suggested to promote protein aggregation and loss of enzyme activity upon long incubation at 37˚C [48]. To examine whether the tails have an impact on the stability of TDG, we kept TDG$^{cat}$ and TDG$^{FL}$ in the reaction buffer at 37˚C for up to 48 h before adding a $^{32}$P-labelled 24-mer 10–13 T*•G duplex (a conventional TDG substrate) and incubating for another 1 h at 37˚C to measure the remaining activity. As shown in Fig 4A and 4B, both truncated and native variants of TDG lost 90% of their activity after 12 h under these conditions (Fig 4A, lanes 4 and 11). Of note, TDG$^{FL}$ exhibited somewhat higher stability with 13% and 3% activity remaining after 3 h and 12 h,

respectively, while TDG$^{cat}$ retained only 9.7% and 2.0% of its activity at these time points (lanes 10–11 *vs* lanes 3–4).

Previously, Maiti *et al.* reported that addition of non-specific DNA prevents TDG from inactivation at 37˚C for 2 h [48]. Therefore, we verified whether the presence of a regular DNA duplex would maintain the enzymes' catalytic proficiency at 37˚C. Adding 20 nM unlabelled 24-mer 10–13 duplex to 200 nM TDG$^{FL}$ or TDG$^{cat}$ slightly stabilized both variants (compare Fig 4C, lanes 3–7, 10–14 and Fig 4D with Fig 4A, lanes 3–7, 10–14 and Fig 4B). Increasing the duplex concentration to 200 nM (i. e, equimolar with the enzyme) did not affect the stability of TDG$^{cat}$, as its activity quickly dropped after 3 h and was completely lost after 24 h at 37˚C (Fig 4E, lanes 3 and 5). Unexpectedly, supplementing 200 nM TDG$^{FL}$ with equimolar unlabelled 10–13 regular DNA duplex dramatically prolonged the enzyme survival at 37˚C (Fig 4E, lanes 9–14, Fig 4F). TDG$^{FL}$ maintained full activity after 3 h at 37˚C and was able to excise 12% of mismatched T even after a 48-h pre-incubation (Fig 4E, lanes 10 and 13). Overall, these results suggest that the presence of a non-specific DNA duplex preserves the catalytic proficiency of a significant fraction of native TDG but not TDG$^{cat}$ in the reaction.

## Single-stranded oligoribonucleotides can modulate the futile activity of TDG

Recently, it has been shown that TDG binds G-rich single-stranded RNAs with high affinity. As a result, short and long RNA molecules with specific sequences can inhibit TDG-catalysed excision from U•G and T•G mismatches in duplex DNA [58]. We examined the effect of short RNA oligonucleotides with varying sequences taken from [58] on the cleavage of 24-mer 10–05 T*•G and T*•A duplexes by TDG$^{FL}$. Consistent with the literature, RNAs unable to form stable complexes with TDG, such as a G-rich RNA hairpin (GC-RNA, Table 1) or single-stranded A-rich RNA (A30-RNA, Table 1) [58], had little or no effect at 0.5–1.0 µM on the G/T-specific TDG activity (Fig 5A, lane 2 *vs* lanes 3–4 and 7–8; Fig 5B). On the other hand, the presence of 0.5–1.0 µM modified G-rich RNA hairpin (Mut-RNA) containing an internal loop and a 3′-single-stranded tail significantly inhibited TDG activity on the T*•G duplex down from 95% DNA cleavage in the control to 68–74% in the presence of RNA (Fig 5A, lane 2 *vs* lanes 5–6; Fig 5B). As shown in Fig 5C, the presence of 0.5–1.0 µM high-affinity Mut-RNA also strongly inhibited futile excision of T from a 24-mer 10–05 T*•A duplex down from 54% in the control to 40% and 12% in the presence of 0.5 µM and 1 µM RNA, respectively (Fig 5C, lane 2 *vs* lanes 5–6). Mut-RNA inhibited the futile activity of TDG to a higher degree than its activity on a T•G mismatch, perhaps because of the differences between TDG binding affinities for RNA *vs* specific and non-specific DNA. Comparing published TDG binding constants for a T•G mismatch ($K_d$ = 18 nM), a T•A duplex (290 nM), Mut-RNA (140 nM), and GC-RNA (2900 nM) [58, 68], one can see that the affinity of TDG for Mut-RNA is 8-fold lower than for the T•G duplex but 21-fold higher than for the T•A duplex. Therefore at the same concentration Mut-RNA will more strongly compete with TDG futile activity than with its G/T mismatch activity. Interestingly, unlike Mut-RNA, low-affinity GC-RNA and A30-RNA moderately stimulated TDG futile activity, up from 54% cleavage of a 24-mer 10–05 T*•A duplex in the control to 67–71% in the presence of RNA (Fig 5C, lane 2 *vs* lanes 3–4 and 7–8). These results suggest that non-specific, low-affinity binding of native TDG to dsRNA and G-poor ssRNA may stabilize the protein conformation and thus increase its stability at 37˚C, similar to what was observed with non-specific DNA duplexes. The presence of an equimolar concentration of unlabelled non-specific DNA did not influence the futile activity of TDG in a statistically significant manner (Fig 5D).

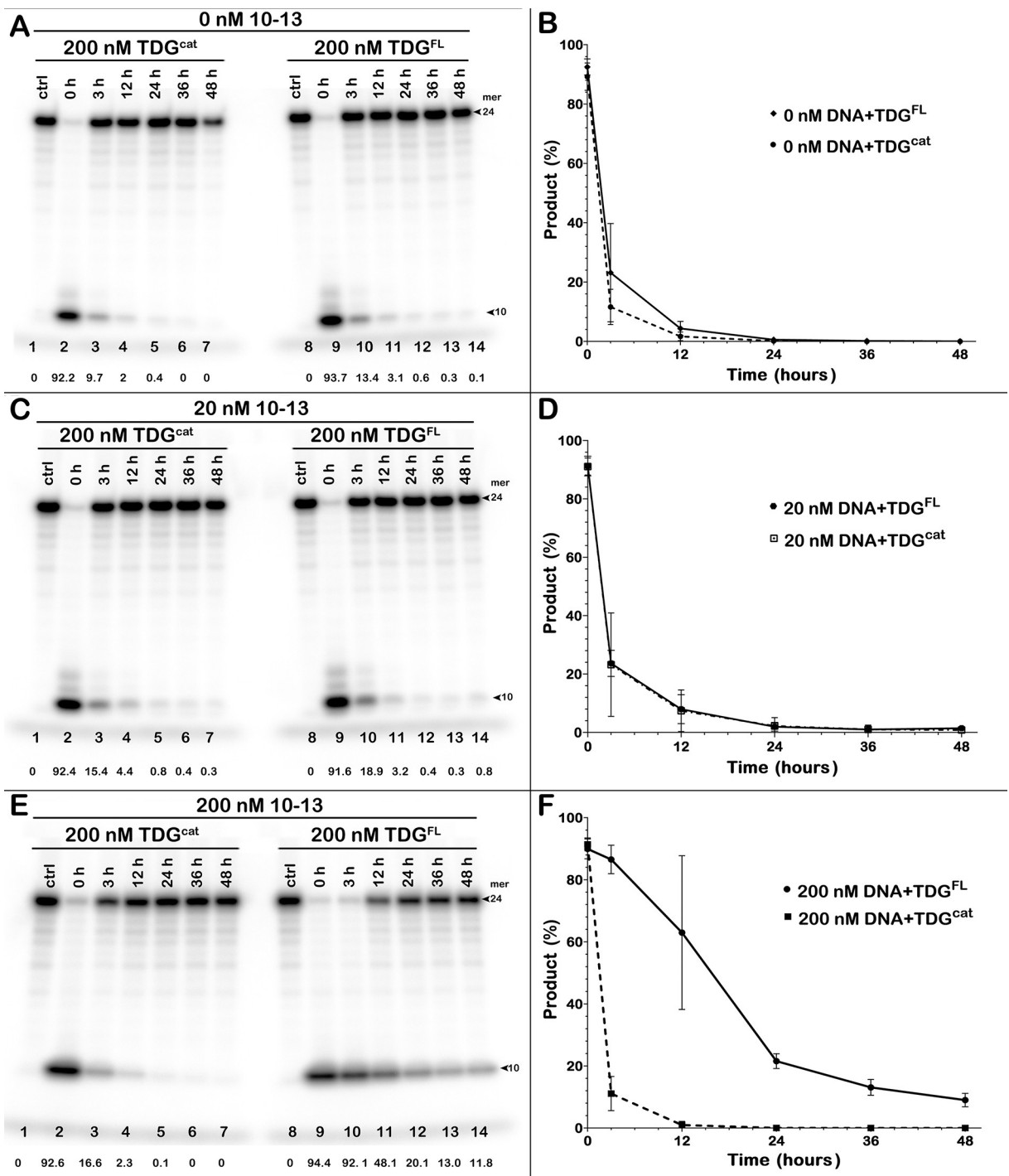

**Fig 4. Loss of the G/T-mismatch-specific DNA glycosylase activity of TDG^cat and TDG^FL upon their prolonged incubation at 37°C.** (**A, C, E**) Analysis of the TDG-catalysed cleavage products of the 24-mer 10–05 G•T* duplex by denaturing PAGE. Prior to the activity assay, 200 nM TDG was incubated at 37°C in the presence of (**A**) 0 nM (**C**) 20 nM, or (**E**) 200 nM non-labelled regular 10–13 DNA duplex. Percentage of cleavage products is indicated under the gel images. Arrows mark the size of the DNA substrate and the cleavage product. (**B, D, F**) Plots of the remaining G/T–DNA glycosylase activity of TDG^cat and TDG^FL pre-incubated in the presence or in the absence of the non-labelled regular duplex. Mean ± SD of three independent experiments is shown. For details, see Materials and Methods.

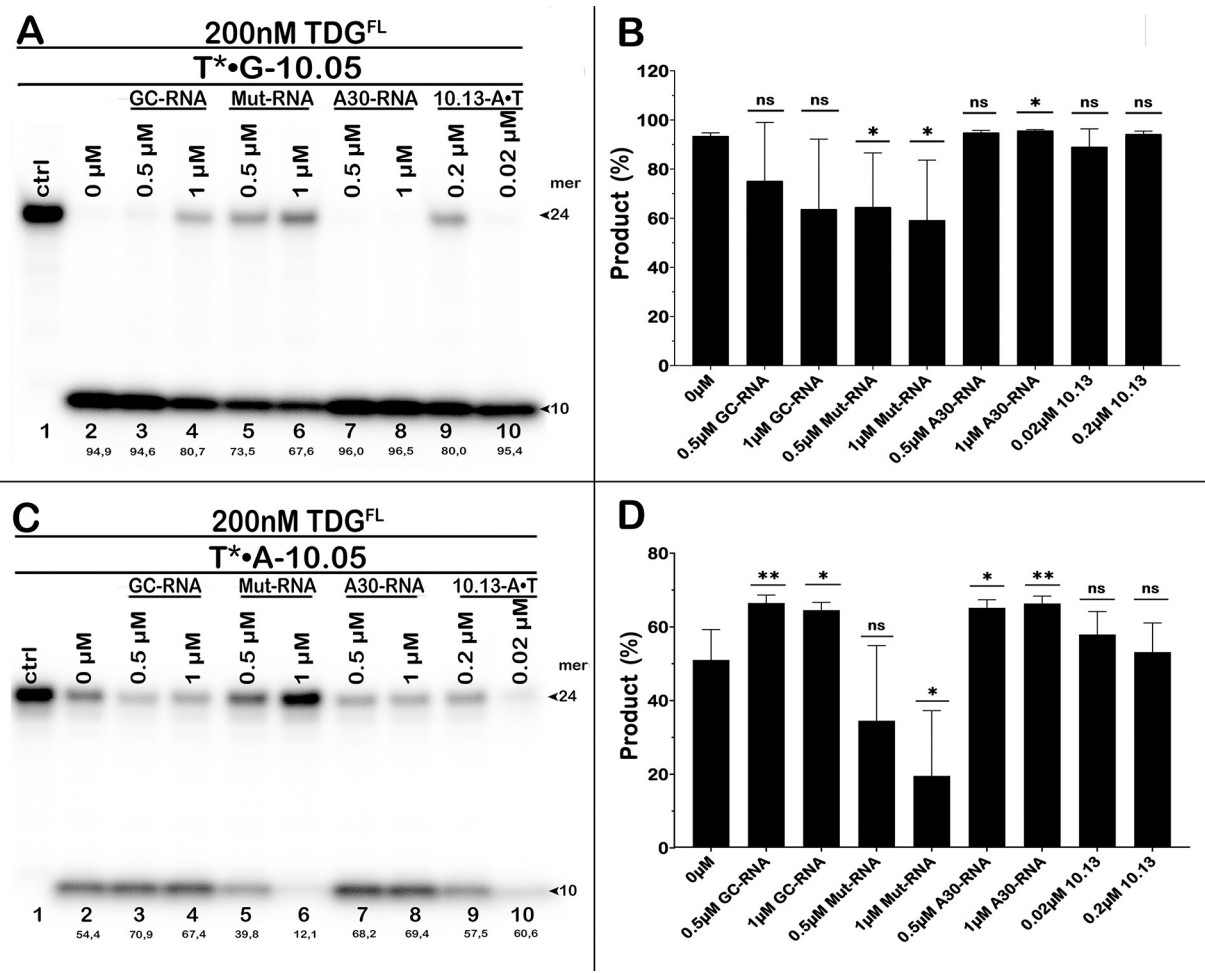

**Fig 5. Impact of RNA oligonucleotides on the G/T-mismatch-specific and futile DNA glycosylase activities of TDG^FL.** (**A, C**) Analysis of the TDG-catalysed cleavage products of 24-mer 10–05 G•T* (**A**) and A•T* duplex (**C**) by denaturing PAGE. Percentage of cleavage products is indicated under the gel images. (**B, D**) Plots of the amounts of the products of cleavage of 10–05 G•T* (**B**) and A•T* (**D**) duplexes. Statistical significance of the differences with the reaction in the absence of RNA was evaluated using two-tailed Student's *t*-test (ns, not significant; *, $p < 0.05$; **, $p < 0.01$). For details, see Materials and Methods.

## Structural basis of TDG stabilization by non-specific nucleic acids

Intrigued by the ability of supposedly disordered N- and C-terminal tails of TDG to entail stabilization on the protein upon interaction with DNA, we explored possible reasons for this phenomenon. To figure out if there might be any communication between the tails, we have analysed the coevolution of amino acid properties in these regions. Correlated changes in physicochemical properties of residues located far apart in the sequence often reflect functional interactions or dynamic physical contacts between them, which may be not evident from a static structure [69, 70]. Taking all TDG homologs from chordates and discarding those lacking the tails, we have used the CRASP tool [60] to search for correlations in the isoelectric point, local flexibility, hydrophobic index (Fig 6A–6C) and several other parameters reflecting hydrophobicity/hydrophilicity, chain flexibility, surface accessibility and tendency to form secondary structure (S6 Fig). As expected, no coevolution could be observed within the TDG catalytic core due to a high level of conservation and a small number of amino acid changes. The C-terminal tail was also almost free of either internal or cross-domain correlations. This is

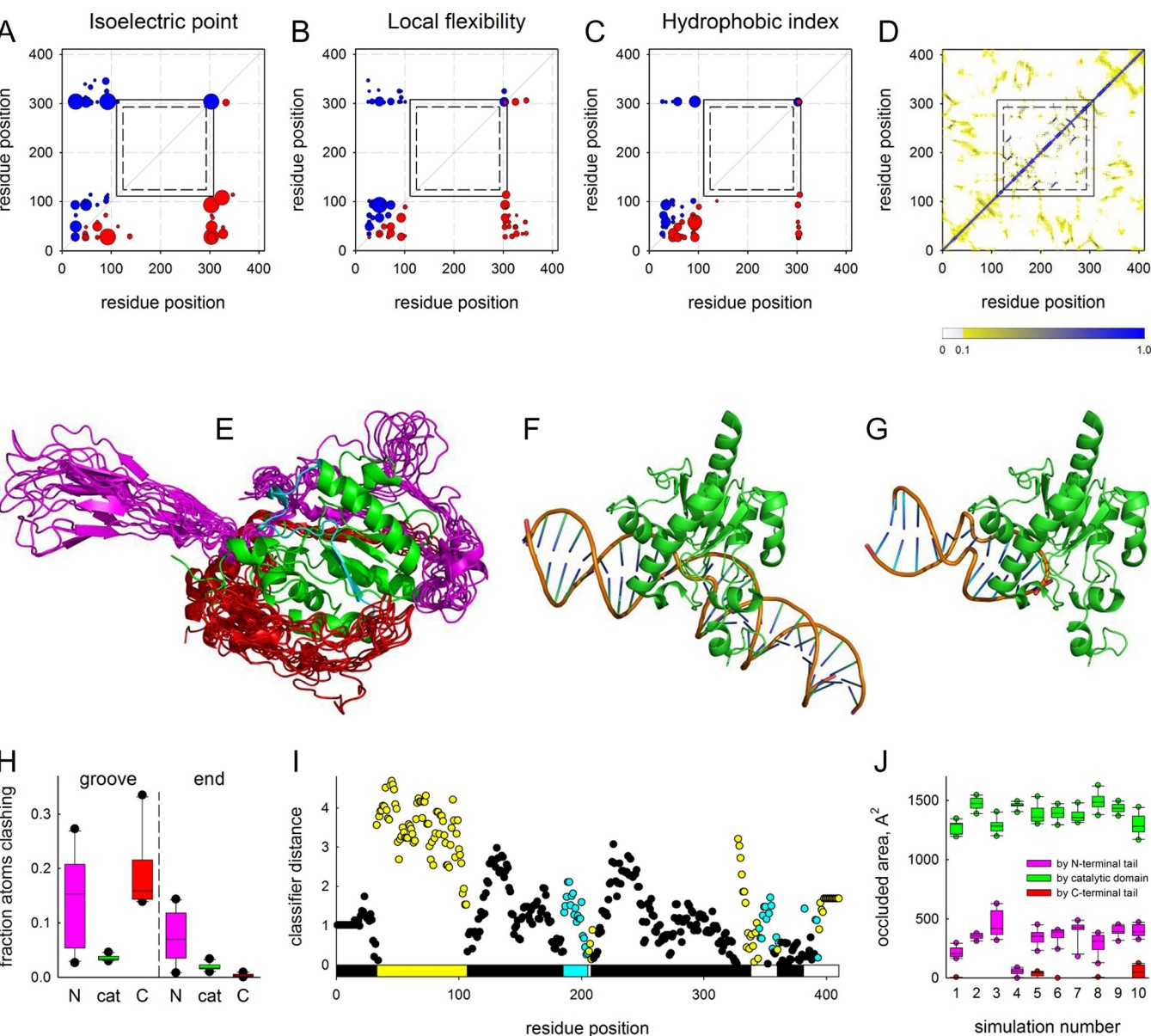

**Fig 6. Analysis of structure and properties of TDG disordered tails.** (**A–C**) Correlation between the isoelectric point [73], local flexibility [74], and hydrophobic index [75] of TDG residues. Red circles under the diagonal, positive correlation; blue circles above the diagonal, negative correlation; circle radii are proportional to the absolute value of correlation coefficient ($r_{ij}$) for the property for the given position pair. Only the pairs with $p < 0.0001$ ($|r_{ij}| > 0.1947$) are shown. The rectangles delimit the catalytic domain [44] (residues 111–308; solid lines) and the UDG-F2_TDG_MUG conserved domain as defined in the NCBI Conserved Domain Database [76] (residues 124–293; dashed lines). (**D**) Color-coded contact frequency map of TDG residues averaged over all coarse-grained simulations. (**E**) Ten centroid structures from one coarse-grained simulation illustrating conformational variability of the TDG tails. The TDG$^{cat}$ domain is shown in green as a single experimental structure (PDB ID 5HF7 [46]); the possible phase separation-prone fragment (residues 186–205) is colored cyan. The cores of the simulated structures are superimposed over TDG$^{cat}$ but omitted from the representation for clarity, leaving only the N-terminal tail (magenta) and the C-terminal tail (red). (**F, G**) Positions of TDG$^{cat}$ bound to the DNA minor groove (**F**) and to the duplex end (**G**); the latter is obtained by superimposing TDG$^{cat}$ over the structure of Mug bound to the DNA end (PDB ID 1MTL, chain A [77]). (**H**) Box plots showing the steric clashes between DNA and the N-terminal tail (N, magenta), TDG$^{cat}$ (cat, green) and C-terminal tail (C, red) of the simulated structures docked in the groove and end positions. Clashes were counted as occurrences of a protein atom within 2.5 Å of any DNA atom. (**I**) ParSe predictions for TDG: cyan, residues intrinsically disordered and prone to undergo phase separation; yellow, intrinsically disordered but do not undergo phase separation; black, may or may not be intrinsically disordered but can fold to a stable conformation; white, no prediction. Bar shows TDG fragments with disorder predictions assigned based on the presence of 20 or more contiguous residues that are at least 90% of the same type. (**J**) Area of the 186–205 fragment occluded by the N-terminal tail (magenta), TDG$^{cat}$ (green) and C-terminal tail (red) of the model centroid structures in individual simulations.

perhaps not surprising since the C-terminal tail provides a wide-area interface for sumoylation, which regulates many of TDG activities [44, 71, 72], and thus might coevolve instead with SUMO proteins or SUMO E3 ligases. However, multiple coevolving residue pairs were observed within the N-terminal tail, which also often showed correlations with residues at the junction of TDG$^{cat}$ and the C-terminal tail (Fig 6A–6C and S6 Fig). This suggests that the N-terminal tail might be at least partially ordered and involved in some structural and/or functional cross-talk with other TDG regions.

At the next step, we sought to probe the conformational space accessible to full-length TDG. To do this, we started with an AlphaFold-generated model, in which both tails are present as extended chains with low confidence score (S7A Fig). The catalytic core is modeled by AlphaFold with a good confidence, with r.m.s.d. = 0.393 over the backbone atoms from the X-ray structure (PDB ID 5HF7 [46]). Since regular full-atom molecular dynamics is poorly suited to sample the huge number of conformations available to long unfolded protein chains, we resorted to coarse-grained Monte Carlo simulations. We used CABS-flex, a modeling tool that implements a knowledge-based force field optimized to provide the best convergence with molecular dynamics of proteins in explicit water at 300 K [64, 65]. Ten independent simulations with random seeds were run, and 1000 models were extracted and clustered for each one. Expectedly, the catalytic core was quite stable during the simulation (r.m.s.d. from the starting model 2.3–3.3 Å) whereas the tails coalesced from the fully unfolded state into better-defined structures, which still demonstrated much higher conformational freedom than the core (Fig 6D, 6E, and S7B–S7K Fig). Interestingly, antiparallel hairpin-like structures were often observed in the tails, especially the N-terminal one (Fig 6D, 6E, and S7B–S7K Fig). Superimposing the models over the structure of TDG$^{cat}$ bound to substrate DNA in the minor groove (Fig 6F) [46] clearly reveals that a substantial conformational rearrangement of both tails would be required to avoid clashing with DNA (Fig 6H). It is known, however, that Mug, the *E. coli* homolog of TDG lacking the tails, can bind ends of a DNA duplex in a non-productive mode [77]. When we docked the models to the position of Mug occupying a DNA end (Fig 6G), it significantly relieved clashes by the N-terminal tail ($p < 0.005$, Student's paired *t*-test) and completely abolished them for the C-terminal tail (Fig 6H). In the former, the atoms escaping from the clash were now more frequently found at distances between 2.5 Å and 4 Å from DNA ($p < 0.05$, Student's paired *t*-test), which is permissive for binding interactions. Also, the hairpin-like structures in the N-terminal tail in many cases could be fit into DNA grooves with slight conformational adjustment.

Some, but not all intrinsically disordered protein regions are prone to undergo liquid–liquid phase separation, which is crucial for spatial organization of many processes, including DNA repair, inside the cell [78, 79]. TDG was reported to induce phase separation and condensation in chromatin both *in vitro* and in cells' nuclei [80, 81]. To see whether the disorder in the TDG structure may assist phase separation, we used ParSe, a family of physics-based classifiers trained on a set of intrinsically disordered proteins to distinguish between condensate-forming and non-separating polypeptide regions [61]. While both tails were predicted to contain disorder-promoting residues of both types, only the N-terminal tail carried a contiguous stretch of such residues long enough to be classified as intrinsically disordered but not prone to phase separation (Fig 6I and S8 Fig). Interestingly, both the basic ParSe algorithm and two extended classifiers with more physicochemical parameters considered robustly identified a fragment of TDG$^{cat}$ (residues 186–205) as intrinsically disordered and prone to undergo phase separation. This region of the core was among the least stable in the simulations (r.m.s.f. plots in S7B–S7K Fig) and, even more intriguingly, is partly disordered in the structure of SUMO3-conjugated free TDG [72]. In the models, the 186–205 fragment contacts the N-terminal tail but not the C-terminal one (Fig 6J), raising the possibility of induced disorder

that may trigger a conformational change in TDG$^{cat}$. Moreover, this fragment also interacts with the end of the duplex in the Mug-like binding mode.

The modeling and bioinformatic exercises allow us to propose a mechanism of TDG$^{FL}$ protection by non-specific DNA. We suggest that TDG$^{FL}$ binds ends of a DNA duplex (or, perhaps, even single-stranded DNA or RNA). In the free state, the conformationally relaxed, disorder-prone N-terminal tail might eventually induce disorder in the 186–205 fragment, possibly followed by TDG condensation. The core of isolated TDG$^{cat}$ where the 186–205 fragment is directly exposed to solution, might also be prone to disorder. However, in the unproductive binding mode, the N-terminal tail is partially diverted to interactions with DNA whereas the 186–205 fragment is protected by its contacts with the duplex end. Thus, while the observed TDG$^{FL}$ protection by non-specific DNA is clearly an *in vitro* phenomenon, it could reflect physiologically pertinent features of TDG related to order/disorder transitions in this important epigenetic regulator and guardian.

## The presence of G•T and G•U mispairs in DNA inhibits futile excision of neighbouring pyrimidines by TDG

TDG is known as a slow-turnover enzyme that lingers at the AP site left after the mismatched base excision for hours and has to be stimulated by APE1, NEIL1 glycosylase or sumoylation to facilitate product release [82–85]. Therefore, we decided to investigate whether the pattern of futile action is affected in the presence of T•G and U•G mispairs in the middle of a long 63-mer oligonucleotide duplex. As shown in Fig 7, 18-h incubation of $^{32}$P-labelled regular 63-mer T*•A and T•A* duplexes with TDG$^{FL}$ generated multiple cleavage fragments (55-, 44-, 38-, 20-, 16-, 13- and 7-mers in the T*•A and 57-, 54-, 45-, 29-, 25-, 23-, 10- and 7-mers in the T•A* substrate) which can be mapped to sites of pyrimidines (T and C) excision from TpG, CpA, CpG and TpA contexts (Fig 7, lanes 2 and 4). As expected, 63-mer T*•G duplex was cleaved by TDG$^{FL}$ in a specific manner, producing a major 31-mer band (Fig 7, lane 6). Consequently, longer futile excision products such as 55- and 44-mer fragments disappeared (since the cleavage is dominated by the mismatched site closer to the labelled 5′-end) while shorter ones (20-, 16-, 13- and 7-mer products) were not affected (Fig 7, lane 6 *vs* lane 2). The generation of short cleavage fragments suggests that tight binding of TDG$^{FL}$ to the AP site product of excision of T opposite to G does not suppress the futile activity at distances of 11 nt or more 5′ of the mismatch. However, when we labelled the complementary strand and compared cleavage of T•G* and T•A* 63-mer duplexes, we observed a pronounced decrease in the generation of the 23-, 25-, 29- and 45-mer, but not of the 7-, 10-, 54- and 57-mer cleavage fragments (Fig 7, lane 8 *vs* lane 4). Thus, the futile excision if strongly inhibited at distances of 14 and 17 nt 3′ of the mismatched G, and 2, 6 and 8 nt 5′ of the mismatched G (Fig 7, lane 8 *vs* lane 4).

In addition, we compared TDG$^{FL}$-catalysed excision patterns using 5′-$^{32}$P-labelled 63-mer U*•G, U•G*, C*•G and C•G* duplexes (Fig 7, lanes 10, 12, 14 and 16). TDG$^{FL}$ excises mismatched U more efficiently than T (Fig 7, lane 10 *vs* lane 6), resulting in a stronger inhibition of the futile activity towards neighbouring pyrimidine bases in a U•G duplex compared to a regular C•G duplex (Fig 7, lane 10 *vs* lane 14 and lane 12 *vs* lane 16). Again, due to the cleavage of U at position 32, all downstream cleavage fragments (54, 44 and 38 nt long) observed with a C*•G duplex disappeared completely (Fig 7, lane 10 *vs* lane 14). Moreover, some upstream products of futile excision such as the 27- and 20-mer cleavage fragments also disappeared completely with the U*•G duplex (Fig 7, lane 10 *vs* lane 14), evidencing suppression of the futile activity on the U-containing strand at distances of 4 and 11 nt 5′ of the mismatched U. In the opposite strand, the products of futile excision 29, 25 and 23 nt long, observed in the C•G* duplex, disappeared completely in the U•G* duplex (Fig 7, lane 12 *vs* lane 16), showing

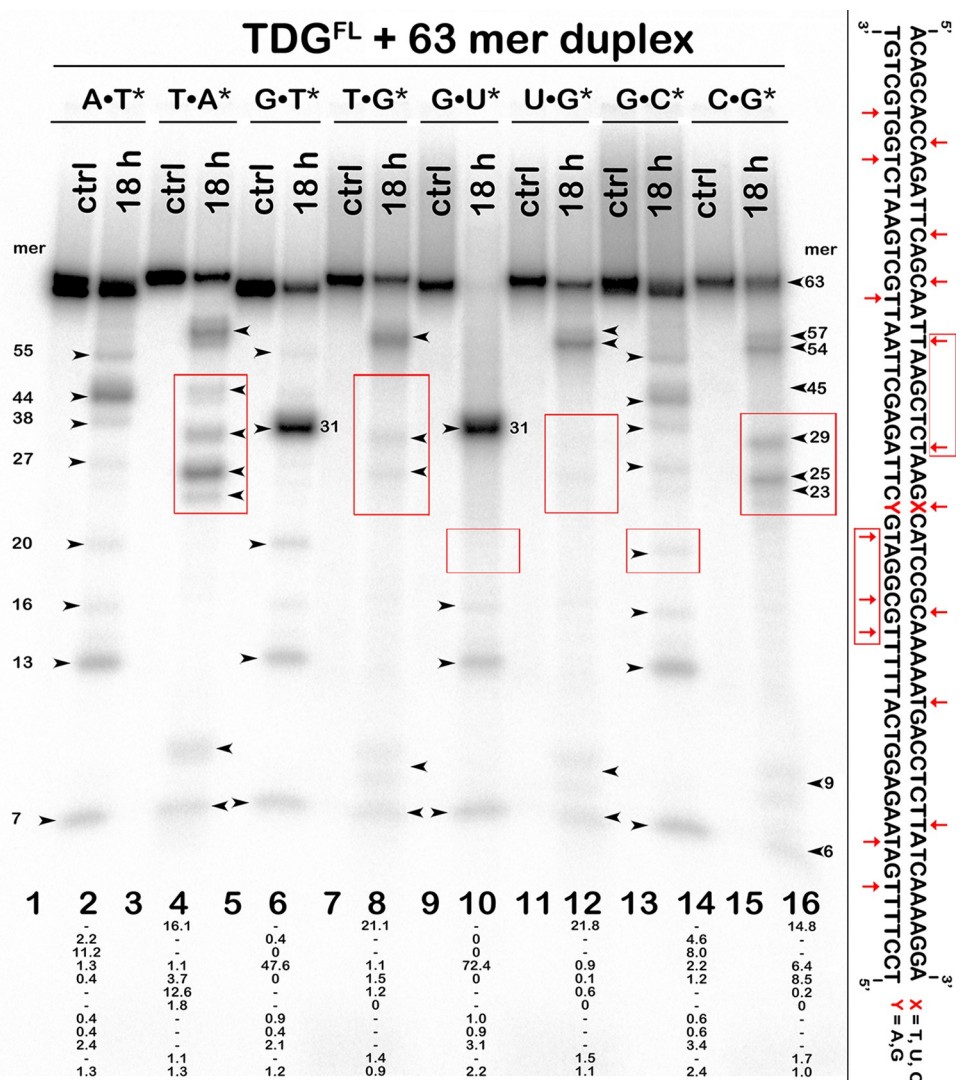

**Fig 7. Action of TDG^FL on 63-mer A\*•T, A•T\* G•T\*, T•G\*, G•U\*, U\*•G, G•C\*, and C•G\* oligonucleotide duplexes in which either the top or the bottom DNA strand is ^32P-labelled.** Denaturing PAGE analysis of the cleavage products is shown. The DNA bands framed in red boxes denote the difference in the cleavage patterns between regular duplexes *vs* T•G and U•G duplexes. Arrows mark the size of the DNA substrate and the cleavage fragments. Percentage of cleavage products is indicated under the gel images. The sequence of the 63-mer duplex is shown right to the gel image with red arrows pointing to the pyrimidines excised by the enzyme. For details, see Materials and Methods.

strong inhibition of the futile activity at the sites 2, 6 and 8 nt 5′ of the mismatched G. At the same time, the presence of a U•G mispair did not suppress futile excision at longer distances, e. g., 15 nt 5′ of U (16-mer cleavage fragment) in the top strand of the U\*•G duplex, and at 23 nt 3′ of the mismatched G (54-mer fragment) and 21 nt (10-mer fragment) 5′ of the mismatched G in the bottom strand of the U•G\* duplex.

Taken together, these results suggest that TDG^FL tightly bound to DNA after highly efficient excision of T and U opposite to G interferes with the TDG^FL-catalysed futile activity in both strands at short distances (8–11 nt) both 3′ and 5′ of T•G and U•G mismatches in duplex DNA. The most obvious explanation is physical interference of the TDG^FL globule residing on DNA with the association of another enzyme molecule nearby. As the physical footprint of

TDG$^{cat}$ on DNA (1 nt 5′ of the mismatch and 3 nt 3′ of the mismatch in the T/U-containing strand; 9 nt 5′ of the mismatch and 4 nt 3′ of the mismatch in the G-containing strand [86]) is shorter than the interference distance, it is feasible that some of the suppression might be effected through the tails of TDG$^{FL}$.

## Futile activity of TDG towards enhancer sequences containing C, 5mC and 5hmC residues in a CpG context

Recently, it has been demonstrated that specific sets of enhancers in post-mitotic neurons are hot-spots for single-strand breaks [27]. We hypothesized that TDG could generate single-strand breaks in enhancers via its futile activity. To examine this possibility, we measured TDG$^{FL}$ activity in two known mouse enhancer DNA sequences, neuronal enhancer element mm876 [56] and ret-inoic acid response element (RARE) involved in the regulation of the *Hic1* (Hypermethylated in Cancer 1) gene [57]. As with other regular DNA duplexes, we observed significant futile excision of T and C in TpG, CpA and CpG contexts from both DNA strands of a 63-mer mm876 duplex and a 27-mer RARE duplex (S9 and S10 Figs). These results may suggest that TDG could intro-duce SSBs in enhancers *in vivo* if stably bound to DNA for a sufficiently long time.

As epigenetic cytosine modifications are common in regulatory DNA sequences, we then investigated whether TDG futile activity can excise the products of cytosine methylation and demethylation, 5mC and 5hmC. Two 27-mer oligonucleotide duplexes, RARE and 14.CpG (an artificial sequence context rich in CpG), containing C, 5hmC, or 5mC at position 15 were used as models. As shown in Fig 8, 18-h incubation of TDG$^{FL}$ with unmodified RARE C*•G duplex generated 14- and 11-mer cleavage fragments corresponding to the excision of C resi-dues at positions 15 and 12, respectively (Fig 8, lane 4; 9.8% at both sites). Interestingly, when TDG$^{FL}$ was incubated with RARE 5hmC*•G and 5mC*•G duplexes for 18 h, the enzyme excised 18% of 5hmC (1.8-fold increase over C) and only 0.8% of 5mC residues (12-fold decrease relative to C) at position 15 (Fig 8, lanes 7 and 10). Similar patterns of excision of C (20%), 5mC (1%) and 5hmC (27%) by TDG$^{FL}$ were observed for the 14.CpG duplex (S11 Fig).

To quantitatively characterise TDG$^{FL}$-catalysed excision of C and its derivatives, we mea-sured the rate constants of base excision ($k_{obs}$) at a defined position in $^{32}$P-labelled 27-mer RARE and 14.CpG T*•G and X*•G (X = C, 5mC or 5hmC) duplexes under single-turnover conditions. As shown in Table 3, the $k_{obs}$ values for excision of C and 5hmC at position 15 opposite to G in the RARE duplex was 110- and 168-fold lower, respectively, than that of mis-matched T excision (0.72 min$^{-1}$). On the 14.CpG substrate, both the regular excision of T and the futile excision of C and 5hmC were ~2-fold higher compared to the RARE duplex, result-ing in similar 118- and 145-fold differences between the G/T mismatch and futile activities in 14.CpG duplexes. Under the conditions used in the single-turnover experiments, in both RARE and 14.CpG sequence contexts TDG$^{FL}$ excised C and 5hmC opposite to G with statisti-cally similar efficiencies (Table 3). We were not able to measure the rate constant of 5mC exci-sion from either duplex because of exceedingly low activity of TDG$^{FL}$ showing less than 1% of cleavage product formation after 18 h of incubation (Fig 7 and S8 Fig). Comparison of the $k_{obs}$ values of TDG$^{FL}$-catalysed futile excision of T, C and 5hmC shows that, despite some sequence context-dependent variation, the rate constants for T•A, C•G and 5hmC•G base pairs were within the same order of magnitude (Tables 2 and 3).

Based on these observations, it is plausible that in post-mitotic cells TDG could generate AP sites further converted to SSBs via the BER pathway in the enhancer DNA sequences con-taining 5hmC residues even without their conversion to 5fC or 5caC. Intriguingly, DNA meth-ylation not only prevents transcription factors binding to enhancers but also strongly inhibits the futile excision of cytosines, thus avoiding appearance of persistent SSBs. It may also follow

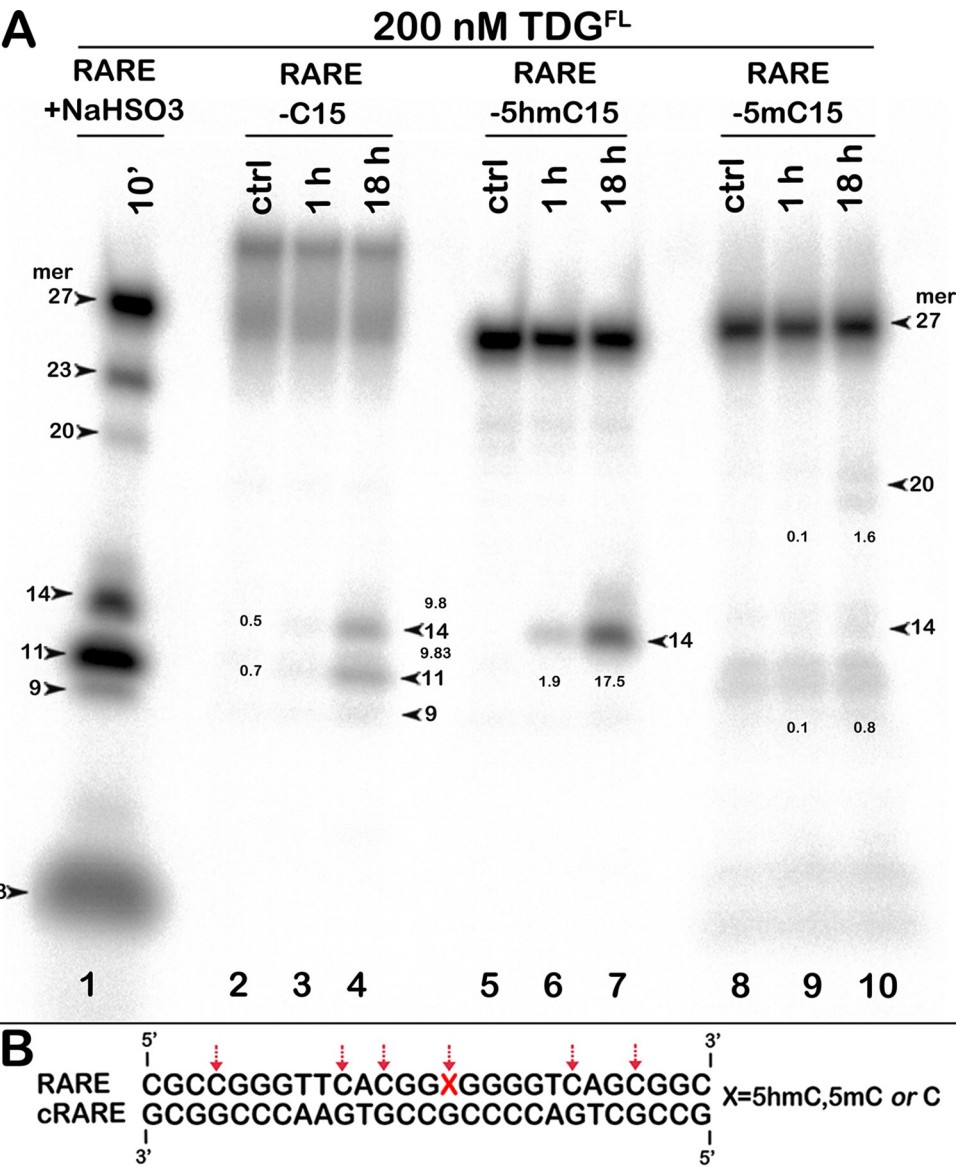

**Fig 8. Action of TDG^FL towards the 27-mer RARE oligonucleotide duplex containing C, 5hmC or 5mC at position 15.** (**A**) Denaturing PAGE analysis of the cleavage products after incubation of TDG^FL with 5′-³²P-labelled 27-mer RARE C*•G, 5hmC*•G and 5mC*•G duplexes for 1 h and 18 h at 37°C. Bisulfite-treated single-stranded 27-mer RARE oligonucleotide was incubated with UNG to generate size markers corresponding to cytosine positions. The numbers next to the bands in lanes 3, 4, 6, 7, 9 and 10 correspond to the percentage of cleavage products. For details, see [Materials and Methods](). (**B**) Schematic representation of the RARE sequence with arrows pointing to the pyrimidines excised by the enzyme.

that TET-mediated generation of 5hmC in enhancer and promoter regions in post-mitotic cells could provide for slow replication-independent DNA demethylation and generation of programmed DNA strand breaks via TDG-initiated BER.

## Discussion

Over the course of evolution, DNA repair systems acquired high specificity for damaged DNA. Nevertheless, challenges to faithful discrimination of lesions among the many orders of

**Table 3. Pre-steady-state kinetic parameters of TDG-catalysed excision of T, C, 5mC and 5hmC opposite to G in 27-mer RARE and 14.CpG duplexes.**

| Substrate[a] | $k_{\text{obs}}$ (min$^{-1}$) | $k_{\text{obs[T]}}/k_{\text{obs[C*]}}$ |
|---|---|---|
| RARE-T15, T15/G13, G•T* (TpG/CpG) | 0.72 ± 0.13 | n/a [b] |
| RARE-C15, C15/G13, G•C* (CpG/CpG) | 0.0065 ± 0.0044 | 110 |
| RARE-5mC15, mC15/G13, G•5mC* (CpG/CpG) | n/d [c] | |
| RARE-5hmC15, hmC15/G13, G•5hmC* (CpG/CpG) | 0.0043 ± 0.004 | 168 |
| 14.CpG-T16, T16/G12, G•T* (TpG/CpG) | 1.5 ± 0.2 | n/a |
| 14.CpG-C16, C16/G12, G•C* (CpG/CpG) | 0.013 ± 0.006 | 118 |
| 14.CpG-5mC16, mC16/G12, G•5mC* (CpG/CpG) | n/d | |
| 14.CpG-5hmC16, hmC16/G12, G•5hmC* (CpG/CpG) | 0.010 ± 0.004 | 145 |

[a] The sequence context is given in the parentheses.

[b] N/a, not applicable for T•G.

[c] N/d, cleavage not detected.

magnitude excess of regular bases in genomic DNA do exist. Under certain conditions, DNA repair can go astray and remove a regular DNA base; the same normal nucleotide is then incorporated, which may initiate multiple rounds of futile repair. Hence, futile base excision repair is useless if not detrimental. In 1998, Seeberg and colleagues discovered that bacterial (AlkA), yeast (MAG) and human (ANPG) alkylpurine–DNA glycosylases can excise regular adenine and guanine bases from non-damaged DNA duplex at measurable rates [40]. Increased spontaneous mutation rates were observed in *E. coli* overexpressing AlkA and MAG, respectively [40, 87]. In humans, an increased level of ANPG is linked with the risk of lung cancer, suggesting that futile BER may contribute to carcinogenesis [41]. Futile DNA repair activity was also shown for the bacterial and human nucleotide excision repair (NER), which can target regular DNA and excise non-damaged oligonucleotide fragments in multiple excision/resynthesis cycles [88]. These futile excision activities of BER and NER are thought to operate primarily on regular DNA, so their mutagenic effect in non-dividing cells may be limited by the error rates of repair DNA polymerases [89]. Intriguingly, when characterizing patterns of somatic mutations in post-mitotic neurons and polyclonal smooth muscle cells, it has been shown that neurons accumulate somatic mutations at a constant rate throughout life without cell division, with rates similar to mitotically active tissues [90, 91]. These observations suggest that mutations generated in the absence of DNA replication and cell division could be an important factor in somatic mutagenesis.

Here, we present evidence that native full-length TDG (TDG$^{\text{FL}}$) can excise regular pyrimidines from T•A and C•G base pairs in duplex DNA in a futile manner with significant efficiency (Figs 1 and 2, and S1 Fig). The rate constants of TDG$^{\text{FL}}$-catalysed excision of T opposite to A in different sequence contexts are only 100–300-fold lower than that of mismatched T in a T•G duplex (Table 2), suggesting that this futile activity could become physiologically significant upon long-time association of TDG with nucleosome-free DNA. Notably, truncation of TDG down to its catalytic domain (TDG$^{\text{cat}}$) lowers the activity on regular DNA duplexes more than tenfold, suggesting that the disordered N- (residues 1–110) and C-terminal (residues 309–410) tails play an important role in the futile activity. MBD4, a functional homolog of TDG, cannot excise pyrimidines from regular DNA, further substantiating the unique mechanism of DNA substrate recognition by the latter (S2 Fig).

At the first glance, our data summarized in Tables 2 and 3 are at odds with previous reports from Drohat's laboratory, where excision of T from T•A base pairs and of C, 5mC and 5hmC opposite to G by TDG, measured at 22–23˚C for up to 50 h, was more than 10$^4$-fold slower

than the excision of T from G•T mispairs [12, 24, 49, 50]. The reason for using the ambient temperature was that TDG irreversibly loses its activity at 37˚C while remaining stable for at least 5 h at 22˚C [48]. The drawback of using the low temperature is that the excision of T by TDG strongly depends on the temperature and is at least threefold higher at 37˚C than at 22˚C even for the G•T mismatch [48]. In agreement with this, our data show that TDG-catalysed futile activity becomes significant only at the physiologically relevant temperature, 37˚C (S5 Fig). Time courses of TDG[FL]-catalysed futile excision of pyrimidines (Fig 3C and 3D) suggest that the native enzyme remains active even after 18 h of incubation at 37˚C. Consistent with the previous work that showed stabilization of TDG in the presence of non-specific DNA [48], we found that the catalytic proficiency of TDG[FL] is greatly stabilized by an equimolar amount of non-specific DNA duplex (Fig 4). The native enzyme kept 100% and 10% of its G/T-specific DNA glycosylase activity after 3 h and 48 h at 37˚C, respectively (Fig 4E and 4F), At the same time, TDG[cat] exhibited a drastic decrease of its G/T-specific activity after 3 h and completely lost it after 24 h at 37˚C, and its stability was not influenced by non-specific DNA. Thus, the dramatic increase in the stability of native TDG in presence of DNA may explain in part the phenomenon of futile DNA repair. In addition, we have shown that low-affinity small RNA oligonucleotides could stimulate TDG[FL]-catalysed futile excision via non-specific binding, which may stabilize protein conformation, whereas high-affinity RNA and DNA with T•G and U•G mismatches inhibits futile activity via strong binding and capturing the glycosylase by RNA and AP sites, respectively (Figs 5 and 7).

Seeking an explanation to the TDG stabilization through interaction with non-specific nucleic acids, we have analysed the interactions and disorder of TDG tails using several bioinformatic and molecular modelling approaches (Fig 6 and S6–S8 Figs). The models suggested that the structure of TDG is highly dynamic, with the N- and C-terminal tails sampling wide conformational space and undergoing rearrangements upon productive DNA binding. The N-terminal tail is presumably engaged in functional cross-talk with the catalytic domain, which could be important for phase separation involving TDG. Unproductive binding to DNA ends likely uncouples the N-terminal tail from TDG[cat], stabilizing the full-length protein.

It is known that G/T-, but not G/U-specific activity of TDG is severely suppressed in DNA packaged into nucleosome core particles [92]. Therefore, our *in vitro* results may suggest that *in vivo* TDG could exhibit futile excision of pyrimidines in open chromatin regions depleted of nucleosomes in post-mitotic non-dividing cells, which would result in the generation of persistent DNA strand breaks. Indeed, we show that *in vitro* TDG is able to excise, at a slow rate, cytosine residues from CpG-rich enhancers mm876 and RARE (S9 and S10 Figs). Intriguingly, recent studies showed that non-dividing neuronal cells accumulate high level of persistent SSBs in the enhancers at CpG dinucleotides associated with TET/TDG-mediated removal of 5mC [26, 27]. It has been suggested that cycles of DNA methylation coupled to active demethylation control the expression of genes involved in cell identity and can be the source of persistent SSBs in differentiated post-mitotic neurons and macrophages [28]. Here, we demonstrate for the first time that native TDG excises 5hmC opposite to G in CpG dinucleotides with better efficiency compared to regular C, whereas 5mC in the same context is resistant to the futile activity (Fig 8 and S11 Fig). Based on these results, we propose that *in vivo* TDG-catalysed BER pathway could participate in slow "active DNA demethylation" of genetic loci rich in 5hmC residues such as enhancers. 5hmC is a stable epigenetic mark typically enriched in transcriptionally active and tissue-specific genes and appears to be important for the maintenance of the cell differentiation status upon division [33–35, 37]. It is tempting to speculate that TDG-mediated direct removal of 5hmC is part of epigenetic mechanisms, which regulate gene expression patterns and preserve specific cellular identities via generation of programmed SSBs.

## Conclusion

Identification and characterisation of TDG-catalysed futile excision of pyrimidines from regular DNA duplexes have significant implications in understanding of the mechanisms of gene regulation and active DNA demethylation. The following major conclusions can be drawn from the present work:

i.  Under the experimental conditions used, TDG catalyses slow sequence context-dependent futile removal of pyrimidine residues from regular DNA duplexes.

ii. TDG futile activity could possibly be involved in the formation of persistent single-strand breaks in quiescent differentiated cells.

iii. Prolonged enzyme stability in the presence of equimolar concentrations of regular DNA duplex suggest that disordered N- and C-terminal domains of TDG can interact with DNA thus preventing formation of insoluble protein complexes, which may explain in part the phenomenon of futile DNA repair.

iv. Our data suggest that 5hmC, similar to 5fC and 5caC residues, can be removed by the BER pathway, albeit at very low rates, and this may be part of epigenetic regulation of tissue-specific gene expression in post-mitotic cells.

## Supporting information

**S1 Fig. The nature of products after TDG$^{FL}$-catalysed cleavage of 41-mer dmbDNA substrates containing either a G•T\* mispair or an A•T\* pair.** dmbDNA with the $^{32}$P-labelled bottom 55-mer strand containing T at position 26 opposite to G (lanes 1–5) or A (lanes 6–10) was incubated with TDG$^{FL}$ for 1 h (lanes 1–5) or 18 h (lanes 6–10) and post-treated with hot alkali (NaOH), light piperidine, or APE1 endonuclease. Lanes 1 and 6, no enzyme; lanes 2 and 7, TDG$^{FL}$, then hot alkali; lanes 3 and 8, TDG$^{FL}$, then light piperidine; lanes 4 and 9, TDG$^{FL}$, then APE1 for 30 min; lanes 5 and 10, TDG$^{FL}$, then APE1 for 1 h. Arrows mark the size of the DNA substrate and the cleavage fragments. For details, see Materials and Methods.
(TIF)

**S2 Fig. Comparison of G/T-specific and futile activities of human mismatch-specific thymine–DNA glycosylases.** Denaturing PAGE analysis of the reaction products generated by TDG$^{FL}$, TDG$^{cat}$ and MBD4 when acting upon 5′-$^{32}$P-labelled 24-mer 10–05 G•T\* and A•T\* duplexes for 1 h and 18 h at 37˚C. Percentage of cleavage products is indicated under the gel images. For details, see Materials and Methods.
(TIF)

**S3 Fig. Time courses of TDG$^{FL}$-catalysed excision of T from G•T and A•T base pairs in the 24-mer 10–05 duplex.** (**A, C**) Denaturing PAGE analysis of the reaction products. Time courses were performed at 37˚C for 0–90 min (**A**) and 0–18 h (**C**) using 20 nM oligonucleotide duplexes where the T-containing strand is 5′-$^{32}$P-labelled. Arrows mark the size of the DNA substrate and the cleavage fragments. Percentage of cleavage products is indicated under the gel images. (**B, D**) Plots of pre-steady-state single turnover kinetic of TDG$^{FL}$-catalysed cleavage of 10–05 duplexes. Mean ± SD of three independent experiments is shown. For details, see Materials and Methods.
(TIF)

**S4 Fig. TDG$^{FL}$ action on regular 28-mer 28Adr and 19-mer AD duplexes in which either top or bottom DNA strand is $^{32}$P-labelled.** (**A, C**) Denaturing PAGE analysis of the reaction

products. Arrows mark the size of the DNA substrate and the cleavage fragments. Percentage of cleavage products is indicated under the gel images. (**A**) Lanes 1–7, 28Adr strand is labelled; lanes 8–12, c28Adr strand is labelled. (**C**) Lanes 1–8, 19AD strand is labelled; lanes 9–14, c19AD strand is labelled (**B, D**) Schematic representation of 28Adr (**B**) and 19AD (**D**) sequences with red with arrows pointing to the pyrimidines excised by the enzyme. For details, see Materials and Methods.
(TIF)

**S5 Fig. Temperature dependence of TDG$^{FL}$-catalysed G/T-specific and futile repair.** 27-mer 14–03 G•T$^*$ and A•T$^*$ duplexes, 28-mer 28Adr/c28Adr and 19-mer 19AD/c19AD A$^*$•T and A•T$^*$ duplexes with either the top or the bottom DNA strand $^{32}$P-labelled were incubated with TDG$^{FL}$ for 1 h or 18 h at 22°C or 37°C. After the reaction, all samples including controls were treated by hot alkali. (**A**) Denaturing PAGE analysis of the products of 14–03 cleavage. Lanes 1–3, 14–03 G•T$^*$ duplex in which the top strand (T at position 12) is labelled: lane 1, no enzyme; lane 2, TDG$^{FL}$ for 1 h at 22°C; lane 3, TDG$^{FL}$ for 1 h at 37°C. Lanes 4–6, same as lanes 1–3 but regular 14–03 A•T$^*$ duplex. (**B**) Denaturing PAGE analysis of the cleavage products of 19AD (lanes 1–6) and 28Adr (lanes 7–12). Lanes 1–3, the 19AD strand is labelled: lane 1, no enzyme; lane 2, TDG$^{FL}$ for 18 h at 22°C; lane 3, TDG$^{FL}$ for 18 h at 37°C. Lanes 4–6, 7–9 and 10–12, same as lanes 1–3 but with c19AD, 28Adr and c28Adr strand labelled, respectively. Arrows mark the size of the DNA substrate and the cleavage fragments. Percentage of cleavage products is indicated under the gel images. For details, see Materials and Methods.
(TIF)

**S6 Fig. Coevolution of TDG tails.** The amino acid properties analyzed for correlations were the overall hydrophobicity (**A, B**) [1, 2], hydrophobicity in folded (**C**) and unfolded form (**D**), hydrophobicity gain upon unfolding (**E**), surrounding hydrophobicity in α-helices (**F**), β-sheets (**H**) and β-turns (**H**) [3], average surrounding hydrophobicity (**I**) [4], hydropathy (**J**) [5], volume (**K**) [6], free energy of transfer from aqueous solution to ethanol (**L**) [7] and to surface (**M**) [8], hydrophilicity obtained from structural (**N**) [9] and HPLC data (**O**) [10], polarity (**P**) [3], overall flexibility (**Q**) [11], flexibility with none (**R**), one (**S**), or two rigid neighbours (**T**) [12], percentage of buried (**U**) and exposed residues (**V**) [13], accessible surface area in the standard state (**W**) and in the folded state (**X**) [14], average accessibility surface area (**Y**) [13], average number of surrounding residues (**Z**), accessibility reduction ratio (**AA**) [3], normalized frequency of α-helices (**AB**), β-sheets (**AC**) and reverse turns (**AD**) [15]. Red circles under the diagonal, positive correlation; blue circles above the diagonal, negative correlation; circle radii are proportional to the absolute value of correlation coefficient ($r_{ij}$) for the property for the given position pair. Only the pairs with $p < 0.0001$ ($|r_{ij}| > 0.1947$) are shown. The rectangles delimit the catalytic domain [16] (residues 111–308; solid lines) and the UDG-F2_TDG_MUG conserved domain as defined in the NCBI Conserved Domain Database [17] (residues 124–293; dashed lines).
(TIF)

**S7 Fig. AlphaFold model of full-length TDG. (A)** and ten models (centroids of the most populated clusters) produced by independent CABS-flex runs (**B–K**). N-terminal tails are colored magenta, TDG$^{cat}$, green, C-terminal tails, red. Arrows point to the hairpin-like structure in the N-terminal tail. Insets show r.m.s.f. profiles along the polypeptide chain, dashed lines delimit the catalytic domain.
(TIF)

**S8 Fig. ParSe predictions for TDG using extended phase separation potential classifiers** [18]. (**A**), classifier additionally trained to consider the standard molar enthalpy associated

with phase separation, $\Delta h^\circ$. (**B**), classifier additionally trained to consider the saturation concentration, $c_{sat}$. The color code is: cyan, residues intrinsically disordered and prone to undergo phase separation; yellow, intrinsically disordered but do not undergo phase separation; black, may or may not be intrinsically disordered but can fold to a stable conformation.
(TIF)

**S9 Fig. TDG$^{FL}$ action on a regular 63-mer mm876 duplex.** (**A**) Denaturing PAGE analysis; 63-mer duplexes in which either the top or the bottom DNA strand is 5′-$^{32}$P-labelled were incubated with TDG$^{FL}$ for 1 h or 18 h at 37˚C. Lanes 1–2, 63-mer T*•G duplex (strands 63 and c63, Table 1); lanes 3–7, mm876 duplex in which the mm876 strand is labelled; lanes 8–12, mm876 duplex in which the c.mm876 strand is labelled. 3′→5′ exonuclease degradation of the mm876 duplexes by Nfo and Xth and TDG cleavage of a 63-mer T*•G duplex were used to generate size markers. Arrows mark the size of the DNA substrate and the cleavage fragments. For details, see Materials and Methods. (**B**) Schematic representation of the mm876 sequence with red arrows pointing to the pyrimidines excised by the enzyme.
(TIF)

**S10 Fig. TDG$^{FL}$ action on a regular 27-mer T*•A RARE duplex.** (**A**) Denaturing PAGE analysis; 27-mer duplexes in which either the top or the bottom DNA strand is 5′-$^{32}$P-labelled were incubated with TDG$^{FL}$ for 1 h or 18 h at 37˚C. Lanes 1–2, 27-mer T*•G duplex; lanes 3–8, RARE duplex in which the RARE strand is labelled; lanes 9–14, RARE duplex in which the c. RARE strand is labelled. 3′→5′ exonuclease degradation of the RARE duplexes by Nfo and Xth and TDG cleavage of a 27-mer T*•G duplex were used to generate size markers. Arrows mark the size of the DNA substrate and the cleavage fragments. Percentage of cleavage products is indicated under the gel images. For details, see Materials and Methods. (**B**) Schematic representation of the RARE sequence with red with arrows pointing to the pyrimidines excised by the enzyme.
(TIF)

**S11 Fig. TDG$^{FL}$ action on 27-mer 14.CpG duplexes containing C, 5hmC or 5mC at position 16.** (A) Denaturing PAGE analysis; 5′-$^{32}$P-labelled 14.CpG C*•G, 5hmC*•G and 5mC*•G duplexes were incubated with TDG$^{FL}$ for 1 h or 18 h at 37˚C. Lane 1, bisulfite-treated single-stranded 14.CpG oligonucleotide was incubated with UNG to generate size markers corresponding to cytosine positions; lanes 2–4, 14.CpG C*•G duplex; lanes 5–7, 14.CpG 5mC*•G duplex; lanes 8–10, 14.CpG 5hmC*•G duplex. Arrows indicate the size of DNA substrate and cleavage products. The numbers next to the bands correspond to the percentage of cleavage products For details, see Materials and Methods. (B) Schematic representation of the 14.CpG sequence with red with arrows pointing to the pyrimidines excised by the enzyme.
(TIF)

**S1 Raw data. Excel file containing raw data for Tables 2, 3 and Figs 4 and 5.**
(XLSX)

**S1 File. Supporting information references.**
(DOCX)

## Acknowledgments

We would like to thank Dr Viktoryia Sidorenko and Arthur Grollman (State University of New York at Stony Brook, Stony Brook, U.S.A.) for their important role in initiating this work.

## Author Contributions

**Conceptualization:** Alexander A. Ishchenko, Bakhyt T. Matkarimov, Dmitry O. Zharkov, Sabira Taipakova, Murat K. Saparbaev.

**Data curation:** Diana Manapkyzy, Botagoz Joldybayeva, Dmitry O. Zharkov, Sabira Taipakova, Murat K. Saparbaev.

**Formal analysis:** Diana Manapkyzy, Botagoz Joldybayeva, Alexander A. Ishchenko, Bakhyt T. Matkarimov, Dmitry O. Zharkov, Sabira Taipakova, Murat K. Saparbaev.

**Funding acquisition:** Alexander A. Ishchenko, Bakhyt T. Matkarimov, Dmitry O. Zharkov, Sabira Taipakova, Murat K. Saparbaev.

**Investigation:** Diana Manapkyzy, Botagoz Joldybayeva, Dmitry O. Zharkov, Sabira Taipakova, Murat K. Saparbaev.

**Methodology:** Diana Manapkyzy, Botagoz Joldybayeva, Alexander A. Ishchenko, Dmitry O. Zharkov.

**Project administration:** Bakhyt T. Matkarimov, Sabira Taipakova, Murat K. Saparbaev.

**Resources:** Alexander A. Ishchenko, Sabira Taipakova, Murat K. Saparbaev.

**Software:** Bakhyt T. Matkarimov, Dmitry O. Zharkov.

**Supervision:** Alexander A. Ishchenko, Sabira Taipakova, Murat K. Saparbaev.

**Validation:** Diana Manapkyzy, Alexander A. Ishchenko, Sabira Taipakova, Murat K. Saparbaev.

**Visualization:** Diana Manapkyzy, Botagoz Joldybayeva, Sabira Taipakova, Murat K. Saparbaev.

**Writing – original draft:** Dmitry O. Zharkov, Murat K. Saparbaev.

**Writing – review & editing:** Dmitry O. Zharkov, Sabira Taipakova, Murat K. Saparbaev.

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
