## [Decision Letter · Decision Letter 0]

12 Jun 2024

PONE-D-24-19416Enhanced thermal stability enables human mismatch-specific thymine-DNA glycosylase to catalyse futile DNA repairPLOS ONE

Dear Dr. Saparbaev,

Thank you for submitting your manuscript to PLOS ONE. After careful consideration, we feel that it has merit but does not fully meet PLOS ONE’s publication criteria as it currently stands. Therefore, we invite you to submit a revised version of the manuscript that addresses the points raised during the review process.

We look forward to receiving your revised manuscript.

Kind regards,

Roshan Thotagamuge, PhD

Academic Editor

PLOS ONE

Journal Requirements:

"This work was supported by grants from the Committee of Science of the Ministry of Science and Higher Education of the Republic of Kazakhstan grants АР 13067762 to S.T. and AP19676334 to S.T. and B.T.M.; French National Research Agency (ANR-22-CE12-0034-01) and Electricité de France RB 2021-05 to M.S.; Fondation ARC PJA-2021060003796 to A.A.I.; Russian Ministry of Higher Education and Science [FSUS-2020-0035] to D.O.Z.; D.M. was supported by fellowship Abai-Vern, Kazakhstan. The funders had no role in study design, data collection and analysis, decision to publish, or preparation of the manuscript."

Additional Editor Comments:

Your manuscript has demonstrated positive merit as noted by the reviewers, but it must undergo major revisions before it can be accepted for publication.

Reviewers' comments:

Reviewer's Responses to Questions

**Comments to the Author**

1. Is the manuscript technically sound, and do the data support the conclusions?

Reviewer #1: Partly

Reviewer #2: No

Reviewer #3: Yes

2. Has the statistical analysis been performed appropriately and rigorously? 

Reviewer #1: No

Reviewer #2: N/A

Reviewer #3: Yes

3. Have the authors made all data underlying the findings in their manuscript fully available?

Reviewer #1: Yes

Reviewer #2: Yes

Reviewer #3: Yes

4. Is the manuscript presented in an intelligible fashion and written in standard English?

Reviewer #1: No

Reviewer #2: No

Reviewer #3: Yes

5. Review Comments to the Author

Reviewer #1: Remarks to the author:

This study presents findings on the role of full-length human thymine-DNA glycosylase (TDG) in the base excision repair pathway. The authors have observed, differential excision rates for various mismatches and the ability of TDGFL to excise 5-hydroxymethylcytosine, but not 5-methylcytosine. Thus, the authors propose that this futile DNA repair could lead to persistent single-strand breaks in non-methylated chromosomal DNA.

Specific comments:

1) The aim/objective of the study is not clearly stated. The authors should clearly mention it in the introductory section.

2) The authors suggest that regular DNA stabilizes the protein through its N- and C-terminal tails, which may bind to the DNA duplex, preventing TDG aggregation and ensuring long-term stability at normal human body temperatures. It would be interesting to understand how the findings of this study apply to the normal physiological environment in the human body.

3) The study lacks a precise conclusion, which is necessary to highlight the significance of the outcomes. Please provide a clear and concise conclusion to emphasize the study's key findings.

4) It would be beneficial if the authors could simplify the content to make it easier for readers to understand.

Reviewer #2: Please see attached reviewer comments for specific comments. As the manuscript stands it requires major revisions in order to support the proposed claims in the manuscript. Furthermore, the naming convention and use of oligonucleotide substrates is sufficiently confusing to this review it makes it challenging to further identify if the author's data supports their claims.

Below are the attached reviewer comments:

Manapkyzy and colleagues have prepared a manuscript entitled Enhanced thermal stability enables human mismatch-specific Thymine-DNA glycosylase to catalyse futile DNA repair. This manuscript suggests that full-length human TDG shows improved enzyme stability as demonstrated by increased activity especially in regards to length of enzyme activity. They further propose that the full-length enzyme is able to excise 5-hydroxymethylcytosine (5hmC), but not 5- methylcytosine residues, and is more active than previously described on T opposite A compared to its preferred substrate T opposite G.

Major comments:

The most contentious argument the authors make is that 5hmC is excised by human TDG and to a lesser extent T:A, and C:G. This has not been previously demonstrated by any group regardless of full-length or partial-length TDG usage. Because of their disagreement with the literature, I feel that the bar to demonstrate this argument is not sufficiently met with the current data provided. If the authors intend to claim this, I think they should include the catalytic rate constants of all proposed substrates, including 5hmC, as they do in Table 2, to determine if other claimed activities are real or not. Furthermore, other groups have used full-length TDG and not seen these activities under similar conditions albeit no incubation for 18 h (1).

This manuscript proposes excision of T opposite A and C excision opposite G. Figure 1 first provides evidence of this, but it is only seen in lanes treated with piperidine but not the bands treated with NaOH. This seems to suggest an artifact and is not sufficiently convincing evidence to me. The authors could consider using APE1 as an additional control lane as it would not require chemical backbone cleavage or reduce radioisotope band intensity.

The naming conventions of their substrates is arbitrary to readers and this reviewer, but only adds confusion. This makes it very challenging to easily interpret their findings and if it agrees with their proposed arguments. I feel this is a major comment as the manuscript is difficult to interpret.

TDG full-length being more active/stable than just the catalytic domain alone has already been described by the Drohat group as well. This is consistent with literature and the Drohat group demonstrated that at least a 29-amino acid portion of the intrinsically disordered N-terminal domain was important for improving activity of the expressed enzyme and reconstituted full-length activity (1, 2). This argument already exists in the literature, and I feel if the authors work also supports this they should describe it in the context of existing literature.

Minor comments:

Figure 2 is non-contributory and interpretation of very minor bands. I do not feel this adds anything to their manuscript.

What is the purpose of using the dumbbell shaped substrate rather than a simple duplex? Why do the authors mix and match fluorescent oligonucleotide data with traditional P32 isotope labeling experiments? Perhaps it may not need to go into the manuscript, but this reviewer is unclear as to some of the experimental design/decisions.

References

1. Baljinnyam, T., Sowers, M. L., Hsu, C. W., Conrad, J. W., Herring, J. L., Hackfeld, L. C., and Sowers, L. C. (2022) Chemical and enzymatic modifications of 5-methylcytosine at the intersection of DNA damage, repair, and epigenetic reprogramming. PLoS One. 17, e0273509

2. Coey, C. T., Malik, S. S., Pidugu, L. S., Varney, K. M., Pozharski, E., and Drohat, A. C. (2016) Structural basis of damage recognition by thymine DNA glycosylase: Key roles for N-terminal residues. Nucleic Acids Res. 44, 10248–10258

Reviewer #3: The introduction states that TDG is known for its wide DNA substrate specificity, excising mismatched DNA substrates, while MBD4 has a narrow DNA substrate specificity. Do the authors base this statement solely on the number of substrates excised by these enzymes? The enzyme activities in prior works have not been compared on a common footing, as those experiments used different initial conditions and setups. It might be better to present this as a conjecture.

Other challenges related to TDG's conformational stability led to a kinetics experiment of BER at 22°C. It is known that non-specific DNA in the reaction buffer stabilizes TDG conformation against heat-induced unfolding at 37°C. This has led to the conjecture that DNA binding prevents protein aggregation, enhances thermal stability, and modulates specificity.

Nonspecific DNA-stabilized TDG catalyzes the aberrant removal of T paired with A, as well as C, 5mC, and 5hmC residues paired with G in non-damaged DNA duplexes. This has been investigated to determine the kinetics of excision. However, it is not entirely fair to presume that specificity can identically discriminate the various derivatives of cytosine at both 22°C and 37°C.

6. PLOS authors have the option to publish the peer review history of their article (what does this mean?). If published, this will include your full peer review and any attached files.

Reviewer #1: No

Reviewer #2: No

Reviewer #3: No

---

## [Author Response · Author response to Decision Letter 0]

1 Aug 2024

ANSWERS TO Referee 1

Specific comments:

Comment 1.1: The aim/objective of the study is not clearly stated. The authors should clearly mention it in the introductory section.

Answer: We thank the Reviewer for highlighting the absence of a clear statement of the aims and objectives in the Introduction. We do agree on this and have updated the explanation of the incentive and the aim of this study at the end of the Introduction section as follows:

“In our previous studies, when characterizing TDG excision of T paired with a damaged adenine, we serendipitously found that upon long incubation at 37°C TDG can also excise pyrimidine bases from control normal DNA duplexes. This led us to inquire into the unusual biochemical properties of TDG acting on regular pyrimidines, 5mC and 5hmC in otherwise undamaged DNA.”.

In addition, we rearranged the Introduction section to put the issue of futile TDG activity in a more logical perspective.

Comment 1.2: The authors suggest that regular DNA stabilizes the protein through its N- and C-terminal tails, which may bind to the DNA duplex, preventing TDG aggregation and ensuring long-term stability at normal human body temperatures. It would be interesting to understand how the findings of this study apply to the normal physiological environment in the human body.

Answer 1.2: We appreciate the Reviewer’s query concerning the importance of conducting in vitro reconstitution assays using physiological conditions. We have performed additional experiments to demonstrate that TDG-catalysed excision of pyrimidines from regular DNA strongly depends on the incubation temperature. For this, we measured TDG-catalysed excision of T from T/A base pairs at 22°C and 37°C. The results show that TDG-catalysed futile repair decreases, depending on the sequence context, more than 5–10-fold at 22°C as compared to 37°C. We have included these data in the revised manuscript as new Supplementary Material Figure S5. This result strongly suggests that normal human body temperature is essential for the behaviour of native TDG and its ability to catalyse the futile base excision.

It should be pointed out that the normal physiological environment in human cells also implies that nuclear DNA is tightly packaged into chromatin, thus severely restraining access to DNA repair proteins. We have already discussed this in the Discussion section of the first version of the manuscript. In the revised version, the pertinent sentences read:

“It is known that G/T-, but not G/U-specific activity of TDG is severely suppressed in DNA packaged into nucleosome core particles (Tarantino et al., 2018). Therefore, our in vitro results may suggest that in vivo TDG could exhibit futile excision of pyrimidines in open chromatin regions depleted of nucleosomes in post-mitotic non-dividing cells, which would result in the generation of persistent DNA strand breaks. Indeed, we show that in vitro TDG is able to excise, at a slow rate, cytosine residues from CpG-rich enhancers mm876 and RARE (S9 and S10 Figs).”

Comment 1.3: The study lacks a precise conclusion, which is necessary to highlight the significance of the outcomes. Please provide a clear and concise conclusion to emphasize the study’s key findings.

Answer 1.3: We agree with the Reviewer’s comment and have now included a new Conclusion section after the Discussion chapter:

“Conclusion

Identification and characterisation of TDG-catalysed futile excision of pyrimidines from regular DNA duplexes have significant implications in understanding of the mechanisms of gene regulation and active DNA demethylation. The following major conclusions can be drawn from the present work:

i. Under the experimental conditions used, TDG catalyses slow sequence context-dependent futile removal of pyrimidine residues from regular DNA duplexes.

ii. TDG futile activity could possibly be involved in the formation of persistent single-strand breaks in quiescent differentiated cells.

iii. Prolonged enzyme stability in the presence of equimolar concentrations of regular DNA duplex suggest that disordered N- and C-terminal domains of TDG can interact with DNA thus preventing formation of insoluble protein complexes, which may explain in part the phenomenon of futile DNA repair.

iv. Our data suggest that 5hmC, similar to 5fC and 5caC residues, can be removed in the BER pathway, albeit at very low rates, and this may be part of epigenetic regulation of tissue-specific gene expression in post-mitotic cells.”

Comment 1.4: 4) It would be beneficial if the authors could simplify the content to make it easier for readers to understand.

Answer 1.4: We thank the Reviewer for this suggestion. We have extensively rearranged and edited the paper to put our experiments and main findings in a more logical order.

ANSWERS TO Referee 2

Comment: As the manuscript stands it requires major revisions in order to support the proposed claims in the manuscript. Furthermore, the naming convention and use of oligonucleotide substrates is sufficiently confusing to this review it makes it challenging to further identify if the author’s data supports their claims.

Answer: Please see Answer 2.4 below.

Major comments:

Comment 2.1: The most contentious argument the authors make is that 5hmC is excised by human TDG and to a lesser extent T:A, and C:G. This has not been previously demonstrated by any group regardless of full-length or partial-length TDG usage. Because of their disagreement with the literature, I feel that the bar to demonstrate this argument is not sufficiently met with the current data provided. If the authors intend to claim this, I think they should include the catalytic rate constants of all proposed substrates, including 5hmC, as they do in Table 2, to determine if other claimed activities are real or not. 

Answer 2.1: We agree with the Reviewer’s advise to include rate constants for TDG-catalysed excision of C, 5mC and 5hmC residues opposite to G in duplex DNA. In the revised version, we have measured the catalytic rate constants for oligonucleotide duplexes with two different sequence contexts containing T/G, C/G, 5mC/G and 5hmC/G pairs at a defined position, which support our initial conclusions. These data have been included into the new Table 3.

Comment 2.2: Furthermore, other groups have used full-length TDG and not seen these activities under similar conditions albeit no incubation for 18 h (1).

Answer 2.2: We agree with the Reviewer’s comment on the apparent discrepancies between our results and the previously published data by Drohat’s group and by Sower’s group (the latter in Baljinnyam et al., 2022). Concerning data from Drohat’s laboratory, we have already discussed in detail (p. 14–15 in the revised version) the possible reasons of the discrepancies, among them the incubation temperature. We have now performed additional experiments to demonstrate that TDG-catalysed excision of pyrimidines from regular DNA duplexes strongly depends on the incubation temperature and the sequence context. We measured TDG-catalysed excision of T from T/A base pairs at both 22°C and 37°C. The results show that, depending on the sequence context, TDG-catalysed futile excision decreases 5- to 10-fold at 22°C as compared to 37°C. This result strongly suggests that normal physiological conditions such as human body temperature are essential for the behaviour of native TDG and its ability to initiate futile repair. We included these data in the revised manuscript in new Supplementary Material Figure S5.

Concerning the data from (Baljinnyam et al., 2022), it should be noted that in this paper, under experimental conditions used, neither TDG nor MBD4 were able to excise T from a T/G mismatch after 1 h, which is in strong contradiction to the majority of studies published on this subject. In many previous studies under similar conditions (10-fold molar excess of enzyme over DNA substrate and incubation at 37°C), both TDG and MBD4 excised ~90% of T/G substrate in 5–60 min (Sibghat-Ullah et al., 1996; Hendrich & Bird, 1998; Abu & Waters, 2003; Hardeland et al., 2003; Bennett et al., 2006). The absence of T/G specific activity in (Baljinnyam et al., 2022) likely indicates that the purified TDG and MBD4 proteins used in that study lost their ability to recognize even some canonical DNA substrates, making them hardly suitable to follow the much slower futile activity.

In the present study, we used fully active TDG and MBD4 enzymes, which were able to exhibit their canonical mismatch-specific thymine–DNA glycosylase activity on the T/G DNA substrate. We used their canonical activity as a control to compare it to futile excision of pyrimidines in T/A and C/G base pairs. It should be emphasized that our data on T/G activity of TDG is similar to that obtained by Drohat’s laboratory and by other researchers.

Finally, to address the Reviewer’s comment, we included new Supporting Figure S5 and have cited the works by Drohat and Sowers in the Introduction section (p. 4, the end of 3rd paragraph in the revised version): “In mammalian cells, erasure of 5mC and 5hmC marks is thought to be strictly dependent on their further oxidation, since no known DNA glycosylase has been reported to excise these nucleobases (Maiti & Drohat, 2011b; Baljinnyam et al., 2022).”

Comment 2.3: This manuscript proposes excision of T opposite A and C excision opposite G. Figure 1 first provides evidence of this, but it is only seen in lanes treated with piperidine but not the bands treated with NaOH. This seems to suggest an artifact and is not sufficiently convincing evidence to me. The authors could consider using APE1 as an additional control lane as it would not require chemical backbone cleavage or reduce radioisotope band intensity.

Answer 2.3: We thank the Reviewer for spotting the difference between light piperidine (LP) and hot NaOH treatments in Figure 1. This difference is simply due to the use of non-freshly prepared oligonucleotides that have less radioactivity because of the natural decay of the 32P isotope. The DNA substrates treated by light piperidine were freshly labelled with new γ[32P]ATP and used immediately for reaction with TDG, whereas the 32P-labelled DNA fragments treated by NaOH were one month old and used in this Figure as additional controls.

To address the Reviewer’s comment in full, we performed an experiment to compare different DNA abasic site cleavage methods side-by-side: we cleaved AP sites resulting from TDG-catalysed base excision with hot alkali (NaOH), LP and human APE1. As shown in the new Supplementary Material Figure S1, hot NaOH generated a fast-migrating 31-mer fragment with a 3′-phosphate terminus, LP treatment yielded the slowest 31-mer fragment with 3′-terminal α,β-unsaturated aldehyde residue, and APE1 produced an intermediate-mobility 31-mer cleavage fragment with a 3′-OH end. We observed no significant difference in the quantity of the products between the treatments, indicating that TDG demonstrates both G/T-specific and futile activities irrespective of the method used to reveal the nascent AP sites. Moreover, the observed differences in the mobility of the post-treated fragments allow unambiguous identification of the TDG futile reaction product as an AP site.

Comment 2.4: The naming conventions of their substrates is arbitrary to readers and this reviewer, but only adds confusion. This makes it very challenging to easily interpret their findings and if it agrees with their proposed arguments. I feel this is a major comment as the manuscript is difficult to interpret.

Answer 2.4: We understand the Reviewer’s concern about the naming of oligonucleotide substrates. It should be noted, though, that the naming of oligonucleotides was based on the previously published studies in which the corresponding sequence contexts were used. For example, the names 14-03, 10-05 and 10-13 were used to designate oligonucleotides containing bulky adenine adducts in (Attaluri et al., 2010; Hashimoto et al., 2016; Bazlekowa-Karaban et al., 2019). The names 28Adr and 19.AD were used to designate oligonucleotides used in Alexander Drohat’s (Adr or AD) laboratory to measure TDG-catalysed activities in (Maiti et al., 2012; Morgan et al., 2007). The names mm876-Enhan and RARE were used to designate sequence contexts of enhancer elements in (Hassan et al., 2017; Song et al., 2019). The names of RNA oligonucleotides were taken from (McGregor et al., 2023). So we kept these names to aid comparison with the data in the literature. Partly to alleviate this mixed-name convention problem, we have included DNA sequence of the duplexes in the figures containing gel images. In the revised version, we edited Table 1 containing the sequences and supplemented it with the appropriate references. Finally, we have revised the naming of oligonucleotide duplexes in Figures 1, 2 and 7.

Comment 2.5: TDG full-length being more active/stable than just the catalytic domain alone has already been described by the Drohat group as well. This is consistent with literature and the Drohat group demonstrated that at least a 29-amino acid portion of the intrinsically disordered N-terminal domain was important for improving activity of the expressed enzyme and reconstituted full-length activity (1, 2). This argument already exists in the literature, and I feel if the authors work also supports this they should describe it in the context of existing literature.

Answer 2.5: We thank the Reviewer for highlighting the works made by our colleagues on the role of the intrinsically disordered flanking domains of TDG in DNA binding and catalysis. To address this comment, we included the following description in the Introduction section (p. 5, 2nd paragraph in the revised version):

“While the observed aberrant activity of TDG is intriguing, functional characterization of this enzyme has long been plagued by the issue of protein stability in vitro. TDG consists of a conserved, well-folded catalytic core and extended, likely disordered N- and C-terminal tails encompassing about a half of the protein’s total length. Multiple studies showed that the N-terminal tail is required for efficient binding and excision of T from G•T mispairs and for binding regular DNA (Gallinari & Jiricny, 1996; Steinacher & Schär, 2005; Guan et al., 2007; Coey et al., 2016). Full-length TDG (TDGFL, 410 amino acids long) and partially truncated TDG (residues 82–308) are severalfold more active than the isolated catalytic domain (residues 111–308, TDGcat) (Coey et al., 2016; Servius et al., 2023). On the other hand, the presence of the intrinsically disordered tails makes TDG quite labile and notably affects its activity and specificity in a temperature-dependent way.”

Minor comment 2.6: Figure 2 is non-contributory and interpretation of very minor bands. I do not feel this adds anything to their manuscript. 

Answer 2.6: We rather disagree with the Reviewer, because Figure 2 demonstrates the nature of TDG-catalysed activity and the role of non-catalytic domains. First, data in Figure 2 clearly show dramatic difference between the full-length TDG and the isolated catalytic domain. TDGFL exhibits 37-fold higher futile excision activity towards T opposite to A in regular duplex DNA as compared to TDGcat. Second, the data shown in Figure 2 indicate that TDG generates abasic sites in DNA via its DNA glycosylase activity. In absence of hot alkali treatment TDG cannot cleave DNA, thus excluding the contamination by non-specific nucleases.

Minor comment 2.7: What is the purpose of using the dumbbell shaped substrate rather than a simple duplex? Why do the authors mix and match fluorescent oligonucleotide data with traditional P32 isotope labeling experiments? Perhaps it may not need to go into the manuscript, but this reviewer is unclear as to some of the experimental design/decisions.

Answer 2.7: We thank the Reviewer for noting the employment of dumbbell DNA conformation and the fluorescent tag for DNA repair assays. As described on p. 11 in the Results section “To avoid degradation of a DNA substrate by a putative nuclease activity, we constructed a 41-mer dumbbell-shaped duplex (dmbDNA)”. Earlier, we used dumbbell DNA substrates in studies of APE1 redox activity for the same end (Bazlekowa-Karaban et al., 2019)

---

## [Decision Letter · Decision Letter 1]

20 Aug 2024

Enhanced thermal stability enables human mismatch-specific thymine-DNA glycosylase to catalyse futile DNA repair

PONE-D-24-19416R1

Dear Dr. Murat ,

We’re pleased to inform you that your manuscript has been judged scientifically suitable for publication and will be formally accepted for publication once it meets all outstanding technical requirements.

Kind regards,

Roshan Thotagamuge, PhD

Academic Editor

PLOS ONE

Additional Editor Comments (optional):

The manuscript has shown improvement after revision and is now suitable for acceptance in the journal.

Reviewers' comments:

Reviewer's Responses to Questions

**Comments to the Author**

1. If the authors have adequately addressed your comments raised in a previous round of review and you feel that this manuscript is now acceptable for publication, you may indicate that here to bypass the “Comments to the Author” section, enter your conflict of interest statement in the “Confidential to Editor” section, and submit your "Accept" recommendation.

Reviewer #1: All comments have been addressed

Reviewer #3: All comments have been addressed

2. Is the manuscript technically sound, and do the data support the conclusions?

Reviewer #1: Yes

Reviewer #3: Partly

3. Has the statistical analysis been performed appropriately and rigorously? 

Reviewer #1: N/A

Reviewer #3: Yes

4. Have the authors made all data underlying the findings in their manuscript fully available?

Reviewer #1: Yes

Reviewer #3: Yes

5. Is the manuscript presented in an intelligible fashion and written in standard English?

Reviewer #1: Yes

Reviewer #3: Yes

6. Review Comments to the Author

Reviewer #1: (No Response)

Reviewer #3: The paper is accepted because the comments raised have been addressed. However some of the concerned points has cleared in author's comments.

7. PLOS authors have the option to publish the peer review history of their article (what does this mean?). If published, this will include your full peer review and any attached files.

Reviewer #1: No

Reviewer #3: No

---

## [Editor Report · Acceptance letter]

28 Aug 2024

PONE-D-24-19416R1 

PLOS ONE

Dear Dr. Saparbaev, 

I'm pleased to inform you that your manuscript has been deemed suitable for publication in PLOS ONE. Congratulations! Your manuscript is now being handed over to our production team.

Kind regards, 

on behalf of

Dr. Roshan Thotagamuge 

Academic Editor

PLOS ONE